# STRUCT2REAL: A SYSTEMATIC FRAMEWORK FOR ACCURATE AND EFFICIENT STRUCTURE-GROUNDED OBJECT IMAGE GENERATION

## ABSTRACT

Recent advances in image generation have enabled the creation of high-quality visual content with impressive semantic fidelity. However, generating object images under fine-grained structural constraints, particularly preserving topology and spatial layout, remains an open challenge. We propose **Struct2Real**, a novel framework for structure-grounded object image generation that combines explicit structural control with photorealistic generation, consisting of twofold. **1)** we develop a novel **structure modeling system** that enables users to create a 3D structural representation named StructMap — an object structure abstraction composed of geometric primitives and their spatial layouts. **2)** We design a modular **image generation algorithm** and combine this algorithm with multimodal large language models (MLLMs), harnessing their superior performance to generate realistic object images under structural constraints encoded in StructMap. Extensive experiments demonstrate that Struct2Real achieves strong performance in structure-grounded object image generation while ensuring low user effort required for this task, highlighting the practicality and effectiveness of our method. *Please refer to more details in our appendix and supplementary material.*

## 1 INTRODUCTION

Image generation has witnessed remarkable progress in recent years, powered by advances in generative models — such as generative adversarial networks (GANs) (Krichen, 2023; Mirza & Osindero, 2014; Isola et al., 2017) and diffusion models (Ho et al., 2020; Song et al., 2021). These techniques have allowed a wide range of applications in creative design (Ruiz et al., 2023), large image synthesis (Le et al., 2024) and story visualization (Wu et al., 2024). In addition to improving image quality, many works (Zhang et al., 2023; Mou et al., 2023; Li et al., 2024; 2025) have explored various input conditions to introduce more control to the generation process, including semantic layouts, sketches, or pose keypoints. These advances reflect the community's growing attention to generating object images that are not only high-quality but also easily manipulable.

Despite recent progress in controllable image generation, producing object images under strong structural constraints remains a significant challenge — particularly when precise control over an object's topology and spatial layout is required. An illustration of these structural constraints is shown in Fig. 2-(a). This level of control is essential because the structural configuration of an object — such as how its parts are arranged, connected, and proportioned — plays a central role in defining its overall shape, category, and functionality. Fine-grained structural control in image generation is crucial for applications such as multiview-consistent image generation (Liu et al., 2023b) and 3D asset design (Tang, 2022), where the topology and spatial layout of objects must be explicitly specified and faithfully preserved. Moreover, structurally controllable and photorealistic images are also crucial for the robotics domain (Tremblay et al., 2018a;b), as the topology and spatial layout of objects are closely related to robotic manipulation.

To effectively generate photorealistic object images under topology and spatial layout constraints, two key challenges need to be addressed. 1) **How to enable users to express their structural intent in an accurate, flexible, and user-friendly manner**. Specifically, topology and spatial layout constraints are difficult to express through conventional input conditions, such as text, semantic

Figure 1: Visualizations of Struct2Real generation results. Each column shows one example: the top row presents a StructMap input representing the object's structure, and the bottom row shows the corresponding photorealistic image generated by our image generation algorithm.

layouts, or pose keypoints, which typically provide only coarse-grained guidance of structural constraints. More fine-grained conditions, such as sketches, while offering greater structural precision, are often difficult for non-expert users to create, as they require both drawing expertise and precise spatial reasoning. 2) **How to design an algorithm that can faithfully translate the structural intent into photorealistic images.** Most existing controllable generation methods such as Control-Net (Zhang et al., 2023; Li et al., 2024) and LoRA (Hu et al., 2022; Dettmers et al., 2023) often perform semantic-level or pixel-level alignment, which hinders them from generating natural geometric details and smooth connections between primitives, making it difficult to simultaneously ensure both realism and structural fidelity in the generated images.

To address these challenges, we propose **Struct2Real**, a novel framework that generates photorealistic object images while preserving topology and spatial layout conditions.

Our method is built on two innovative core designs. **First, we introduce an effective 3D representation for object structure to the image generation task and design a structure modeling system around it.** This 3D representation encodes the topology and spatial layout of an object using a composition of geometric primitives. We refer to this 3D structural representation as StructMap. The structure modeling system ensures that users can express their structural intent accurately, flexibly, and conveniently, while providing an expressive prior for controllable image generation. **Second, we design a novel image generation algorithm building on the strong capabilities of multimodal large language models (MLLMs) in understanding structural and semantic information in images and generating high-quality visual content.** The algorithm is able to generate high-quality images while maintaining strong structural fidelity, even in challenging scenarios involving complex topologies and fine-grained part relations, without requiring any task-specific training or fine-tuning. Examples of StructMaps and corresponding generated object images are shown in Fig. 1.

We compare Struct2Real against state-of-the-art baselines on generating photorealistic object images under topology and spatial layout constraints. The comparison covers multiple evaluation aspects, including image realism, structural alignment, and condition accessibility. The results demonstrate that Struct2Real achieves superior performance in generating photorealistic and structurally faithful images at the cost of reasonable user effort.

In summary, this work makes the following key contributions: 1) In the task of structure-grounded object image generation, we introduce an effective 3D representation called StructMap and develop a StructMap-driven structure modeling system, enabling users to express their structural intent accurately, flexibly, and conveniently. 2) We design an image generation algorithm and combine it together with MLLM to generate photorealistic object images under precise structural control, while exploring the application of various MLLMs in controllable image generation tasks at the same time. 3) We conduct comprehensive experiments and provide in-depth analysis to study the effectiveness of Struct2Real and its components, and the results demonstrate its superiority.

## 2 RELATED WORKS

### 2.1 CONTROLLABLE IMAGE GENERATION

As the most fundamental forms of controllable image generation, text-to-image generation has made significant progress (Nichol et al., 2022; Ramesh et al., 2022; Rombach et al., 2022; Saharia et al., 2022) largely due to the advent of diffusion models (Ho et al., 2020; Song et al., 2021), which

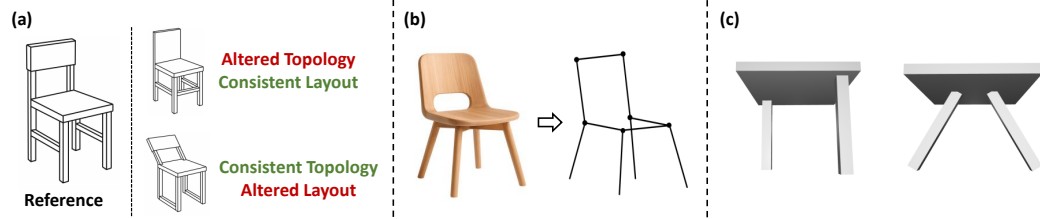

Figure 2: **(a)** Illustration of an object's topology and spatial layout using chair as example. **(b)** An example of how to represent the structure of a chair using points and lines. **(c)** An example illustrating two different ways in which two legs are connected to a seat. This example demonstrates that our representation enables a finer-grained characterization of the spatial relationships and connection between components.

enable high-quality synthesis conditioned on textual inputs. However, these models still fall short in providing controllable image guided by more specific conditions. As a result, many recent approaches have incorporated additional modalities as complementary control signals. For instance, semantically annotated bounding boxes (Cheng et al., 2023; Jia et al., 2024a; Qu et al., 2023; Xie et al., 2023; Yang et al., 2023a;b; Zheng et al., 2023) can help generate images with strong layout alignment, segmentation maps (Avrahami et al., 2023; Bar-Tal et al., 2023; Couairon et al., 2023; Gafni et al., 2022) enable better control over both shape and layout, edge-based inputs such as sketches (Bashkirova et al., 2023; Wang et al., 2024b; Koley et al., 2024) provide highly detailed control, depth maps (Lee et al., 2024) control the depth of field for the generated images, and methods like InstanceFusion (Wang et al., 2024c) and AnyControl (Sun et al., 2024) further unify multiple input types (*e.g.* points, scribbles, boxes, segmentations) into a flexible control interface. Additionally, some works focus on subject-driven (Yang et al., 2024b; Chen et al., 2024; Dong et al., 2024) and style-specified (Han et al., 2025; Zhang et al., 2024; Rout et al., 2025) image generation, allowing for appearance personalization at the semantic level. Despite progress in multimodal conditioning, current methods offer limited fine-grained control over object topology and spatial layout.

## 2.2 MLLM FOR IMAGE GENERATION

Large Language Models (LLMs) have demonstrated strong capabilities in language understanding (Brown et al., 2020) and problem solving (Yao et al., 2023). As LLMs evolve to handle multimodal inputs, Multimodal Large Language Models (MLLMs) have gained increasing attention for their potential to enhance visual content generation. For example, (Feng et al., 2023; Lian et al., 2024; Yang et al., 2024a; Wang et al., 2024e) employ MLLMs to generate detailed layout information, other works (Feng et al., 2024; Tan et al., 2024; Liu et al., 2024; Xia et al., 2024; Song et al., 2024) use LLMs as enhanced text encoders, producing intermediate features that guide traditional generators. Meanwhile, (Qin et al., 2024; Wang et al., 2024d; Zhao et al., 2024; Jia et al., 2024b) treat MLLMs as agents capable of tool selection and orchestration within multi-component generation systems. Beyond these roles, LLMs have also been explored for their reasoning and refinement capabilities. For instance, (Wu et al., 2023) and (Yang et al., 2024c) propose using LLMs to support self-correction during generation. In addition, LLMs have been shown to benefit image generation involving rare concepts (Park et al., 2025), and to improve both global coherence and local consistency (Kwon et al., 2024). These works highlight the growing synergy between language models and image generation. In our work, we design an image generation algorithm to better exploit the capabilities of MLLMs for controllable image generation, and explore the application of various MLLMs in controllable image generation tasks.

## 3 METHOD

### 3.1 OVERVIEW

Our goal is to generate object images under structural control, ensuring that the outputs not only appear natural, realistic, and detail-rich, but also satisfy topology and spatial layout constraints.

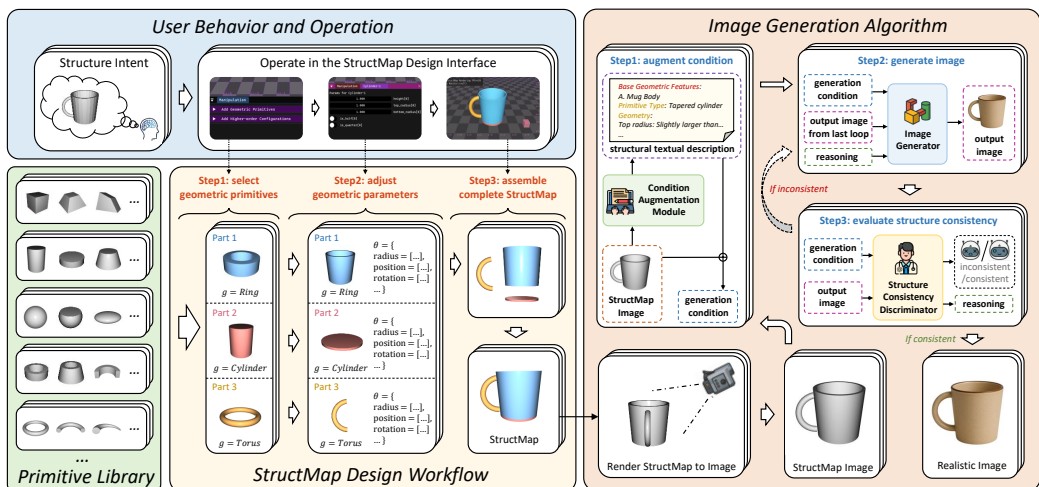

Figure 3: Overview of the Struct2Real pipeline for generating structurally controllable photorealistic object images. The user first creates the desired StructMap according to their structural intent using our StructMap design workflow and interface. The resulting StructMap is then rendered as a StructMap image and provided as the condition for our image generation algorithm, which produces photorealistic images that faithfully follow the structure encoded in the StructMap.

Formally, we need to find a structural condition $S_M$ that encodes the object's topology and spatial layout $S$, and then produce an image $I = \mathcal{F}(S_M)$, where $\mathcal{F}$ denotes a generative process. The generated image $I$ is expected to possess two key properties: (i) **Structural Faithfulness**: preserving the structural configurations defined by $S_M$. (ii) **Visual Realism**: appearing nearly indistinguishable from real-world objects, as if captured in actual photographs.

To this end, we present Struct2Real, a framework with two main modules: 1) a structure modeling system that allows users to conveniently create a novel structural condition $S_M$, called StructMap, which accurately encodes the topology and spatial layout of an object. 2) an image generation algorithm $\mathcal{F}$ that generates photorealistic images $I$, while preserving the structural configurations defined by $S_M$. In the following sections, we introduce these components and the generation process in detail. An illustration of our pipeline is shown in Fig. 3.

## 3.2 STRUCTMAP AND STRUCTURE MODELING SYSTEM

While previous methods for controllable image generation have explored a variety of input conditions such as text, semantic layouts, or pose keypoint, these conditions are often coarse-grained or ambiguous, making it difficult to accurately reflect the object's topology and spatial layout, *which is discussed in detail in Appendix A.1*. And more fine-grained conditions, such as sketches, are often difficult for non-expert users to create, as accurately depicting object structures typically requires both artistic expertise and strong spatial understanding. In comparison, we develop a structure modeling system that enables users to create a effective 3D structural representation named StructMap. It serves as a prior to guide the generation process with explicit control over object topology and spatial layout. Building upon StructMap, we develop a part-based workflow embedded in an interactive design interface allowing users to conveniently organize geometric primitives into StructMaps with fully customizable topology and spatial layout.

**Definition of StructMap.** We define a **StructMap** as an explicit 3D representation encoding an object's topology and spatial layout, composed of a set of geometric primitives and their spatial relationships. Formally, a StructMap is represented as a primitive set $\{(g_i, \theta_i)\}_{i=1}^{N}$, where each $g_i \in \mathcal{G}$ is a geometric primitive selected from a primitive library $\mathcal{G}$, and $\theta_i$ is a set of interpretable parameters that determine the primitive's shape, position, and orientation in 3D space. Each primitive $(g_i, \theta_i)$ serves as a structural component or "part" of the object. The topology of the object is revealed by the contact relationships of these parts, while the spatial layout is encoded by their absolute and relative positions in 3D space. Fig. 3 contains an illustrative example of StrcutMap's definition.

**Motivation and Rationale Behind StructMap.** Our goal is to develop a 3D representation that can flexibly and accurately encode an object's topology and spatial layout, while remaining intuitive and accessible for users to create. The most straightforward approach is to characterize an object's structure using points and lines (Haridis, 2020; Xu et al., 2025), where each line represents a component of the object and each point represents the connections between components, as illustrated in Fig. 2-(b). However, since points and lines have no 3D volume, this representation cannot capture the specific size and shape of each component. Such natural disadvantage makes it difficult to represent certain complex topology and it also prevents precise specification of the connection locations between components — for example, a seat with four legs attached precisely at the quarter points along its diagonals. To overcome these limitations, we propose representing each basic components as a 3D geometric primitive with controllable size and pose, thereby enabling a finer-grained representation of their spatial relationships and connections, such as the example illustrated in Fig. 2-(c), two legs can be attached to the seat at any position and in any manner. Building on prior studies of geometric primitives and our observations of a wide range of real-world objects, we carefully select a set of representative geometric primitives and design shape-controlling parameters for them. This ensures that the primitives can effectively encode the topology and spatial layout of diverse objects while remaining simple and convenient for users to manipulate. Moreover, this design also ensures the extensibility of the representation, as additional primitives and richer parameterizations can be introduced to support expressive and scalable modeling across various object categories.

This representation offers a cognition-inspired approach to model object structure, aligning with how humans perceive and reason about 3D objects. According to classic theories in cognitive science, particularly the 'Recognition-by-Components' theory (Biederman, 1987), humans recognize objects not by memorizing textures and geometric details, but by decomposing them into a small set of simple, geometric primitives along with their spatial configuration. This structural abstraction effectively satisfies the goals of controllable image generation: it enables users to clearly and precisely specify their intent on the structural condition, and supports fine-grained structural adjustments.

**StructMap Design Workflow and Interface.** We develop a part-based workflow tailored for designing StructMaps. This workflow defines how users can organize geometric primitives into meaningful structural configurations. First, users select and instantiate geometric primitives from the primitive library according to their design requirements. Since each primitive can be adjusted via interpretable parameters such as size, position, and orientation, users can modify them to achieve the desired geometry and pose. Once all primitives have been composed and configured, the user completes the construction of a StructMap that explicitly encodes their intended structure. To support this workflow and facilitate practical usage, we implement an interactive design interface that enables users to build StructMaps. In this interface, we design several high-order configurations based on common object structures, enabling users to instantiate multiple primitives simultaneously. *Examples of the high-order configurations are provided in Appendix A.2.* And the interface allows users to reuse previously created StructMaps as components when creating new ones. These designs further accelerate the creation of StructMaps. We include screenshots of our interface in Fig. 3. *A more detailed video demonstration of our workflow and interface is provided in the supplementary material* `design_workflow_demonstration.mp4`. Once a 3D StructMap is created, we render it as a 2D image from one or more viewpoints, forming a structural visual prompt for the subsequent generation. In the meantime, the 3D StructMap can be exported and reused in case of further editing.

**Discussions.** 1) Compared to sketches, StructMap is natively 3D and thus naturally encodes 3D attributes such as depth, angle of view, and symmetry, while also lowering the skill barrier for users. 2) Compared to StructMap, coarse-grained conditions such as text, semantic layouts, and pose keypoints provide only coarse-grained descriptions of an object's structure. 3) Unlike CAD, which involves precise 3D modeling and requires professional design expertise, our system is purpose-built for structure modeling, offering a more accessible and user-friendly alternative. *Please refer to Appendix A.1 for more detailed discussions.*

### 3.3 IMAGE GENERATION ALGORITHM

After creating a StructMap that encodes the user's structural intent, we need to design an algorithm to leverage the comprehensive structural information in the StructMap and generate photorealistic object images while preserving the structural configuration it encodes. We found that MLLMs

possess strong capabilities in understanding structural and semantic information in images and generating high-quality visual content, and motivated by this, we design an image generation algorithm to activate and leverage these capabilities. The algorithm consists of three components — the Condition Augmentation Module, the Image Generator, and the Structure Consistency Discriminator — each assigned a novel functionality, and we further devise tailored prompting strategies to enable them to collaborate effectively. The detailed workflow of our algorithm is presented in Fig. 4, and in the following, we describe the functionality and implementation of each component in detail.

**Condition Augmentation Module.** The condition of our image generation process is a StructMap image, which encodes all necessary structural configurations of the object. However, to reduce the modality gap between the image input and the language input preferred by the MLLM (Liu et al., 2023a), we augment the condition further by translating the StructMap image into a textual description. This textual description complements the image by describing the StructMap's topology and spatial layout, *i.e.* the number of parts, their spatial arrangement, relative scale, and connectivity. Specifically, we design a prompt that introduces the core properties of StructMap to the MLLM by clarifying that it encodes an object's structural properties while omitting surface-level appearance. We then employ the MLLM to perform this augmentation, leveraging its multimodal understanding capabilities to extract and emphasize the structural configurations represented in the StructMap.

**Image Generator.** Conditioned on both the StructMap image and the structural textual description, MLLM serves as an image generator to produce a photorealistic image that faithfully preserves the topology and spatial layout encoded in the StructMap. We explicitly defines the image generation objective in the prompt: to generate a photorealistic object image that preserves the structural configurations reflected in the StructMap, while synthesizing plausible fine-grained visual details including textures, materials, and geometric details. To support further control, our algorithm also allow users to optionally provide a free-form prompt to specify visual attributes, such as material type or aesthetic style, enabling appearance customization without compromising structure.

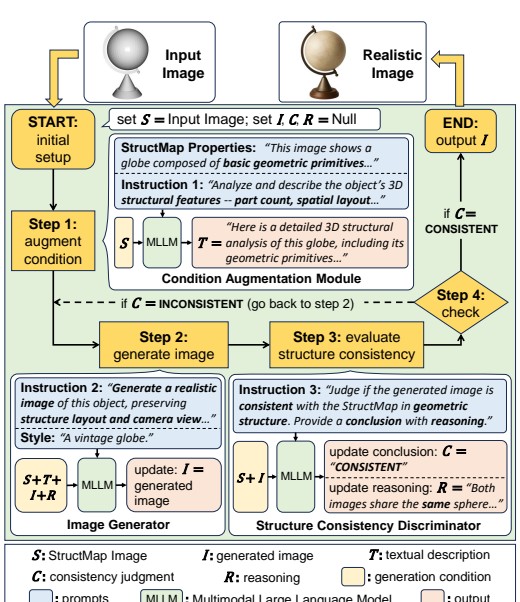

Figure 4: Illustration of our image generation algorithm. *The complete prompt is provided in Appendix A.7.*

**Structure Consistency Discriminator.** In our preliminary experiments, we observed that the image generator occasionally produces results inconsistent with the given structural conditions. To address this problem, we introduce a structure consistency discriminator to evaluate whether the generated object image preserves the structural constraints defined by the StructMap. Specifically, we feed both the StructMap and the generated result back into MLLM and prompt it to compare their structures, identify any inconsistencies, and provide a reasoning. If discrepancies are found, the algorithm instructs the image generator to regenerate the image, while incorporating the previously generated erroneous image and the reasoning as supplementary input to guide a more accurate regeneration. This feedback loop can be repeated iteratively until the discriminator confirms that the generated image matches the StructMap without any remaining structural discrepancies. Notably, to ensure that the algorithm does not fall into an infinite loop, we set the maximum number of iterations to 5, and we show the iteration statistics in practical use in Appendix A.14.

**Discussions.** Our image generation algorithm does not need to be executed strictly following the full pipeline in practical usage. We provide the most rigorous workflow to ensure robustness across the majority of cases, and in certain scenarios, users may decide whether to apply the Condition Augmentation Module and the Structure Consistency Discriminator for efficiency considerations, at the cost of a slight reduction in generation quality.

Table 1: **[Left]** Quantitative comparison across different structural conditions and approaches. **[Right]** Average creation time and creation difficulty (on a 1–5 scale) for each input condition.

| Condition | Method | FID↓ | MOS-R↑ | MOS-A↑ | Creation Time (min) | Creation Difficulty |
|---|---|---|---|---|---|---|
| Text | OmniGen | 43.17 | 1.73 | 1.25 | 2.79 | 2.09 |
| | Lumina-Image 2.0 | 43.49 | 1.67 | 1.42 | | |
| | GPT-4o | 39.29 | 4.33 | 1.92 | | |
| Lineart | ControlNet++ | 41.56 | 2.67 | 4.62 | 25.76 | 4.39 |
| | ControlAR | 49.97 | 1.61 | 4.89 | | |
| | GPT-4o | 38.78 | 4.58 | 4.51 | | |
| Scribble | ControlNet | 40.37 | 2.88 | 3.51 | 3.26 | 2.26 |
| | T2I-Adapter | 42.97 | 2.53 | 3.87 | | |
| | GPT-4o | 38.87 | 4.19 | 4.05 | | |
| StructMap | Struct2Real | **38.61** | **4.65** | **4.56** | 6.69 | 2.68 |

## 4 EXPERIMENTS

We conduct comprehensive experiments to evaluate the effectiveness of Struct2Real. We first describe our experiment setup in Sec. 4.1. Then, we present qualitative and quantitative results and analyses in Sec. 4.2. Finally, we conduct ablation study in Sec. 4.3. *Due to space limitation, more results and analyses are provided in Appendix.*

### 4.1 EXPERIMENT SETUP

**Data Preparation.** Existing datasets for controllable image generation are often constructed by extracting conditioning inputs from pre-existing images (Lin et al., 2015). While this approach is convenient, it limits the input conditions to those derivable from existing images and thereby restricts user's creativity. Moreover, these datasets typically lack alignment across different conditions, making it difficult to evaluate structure-grounded generation in a consistent and comparable way. To address these limitations, we construct a structure-prior dataset comprising diverse, manually created conditioning inputs across multiple conditions, enabling unrestricted condition creation and cross-condition comparison. Among the various types of conditioning inputs commonly used in image generation, we focus on four conditioning inputs that can be flexibly created from scratch by users: textual descriptions, lineart, scribbles, and our proposed StructMap. To ensure fair comparisons across different conditions, we require that each set of conditioning inputs describe the same object structure. Formally, for each structure $S$, we collect a tuple $(T, L, S_C, S_M)$ where $T, L, S_C, S_M$ denote textual description, lineart, scribble and StructMap respectively, and all aimed at representing the same $S$. Overall, our dataset covers 30 distinct object categories and includes a total of 3000 samples. *Additional details on dataset construction are provided in Appendix A.3.1, and we also present the diversity and complexity of StructMap structures in the dataset in Appendix A.10.* Examples of input conditions are shown in the top row of each sub-panel in Fig. 5.

**Baseline Methods.** To comprehensively evaluate our method, we compare against strong baselines under each conditioning input. For each condition, we select two representative models that are either widely adopted or represent state-of-the-art advancements under that condition. Specifically, we compare against OmniGen (Xiao et al., 2024) and Lumina-Image 2.0 (Qin et al., 2025) for text condition, ControlNet++ (Li et al., 2024) and ControlAR (Li et al., 2025) for lineart condition, and ControlNet (Zhang et al., 2023) and T2I-Adapter (Mou et al., 2023) for scribble condition. [1] *Detailed introductions to these baselines are provided in Appendix A.3.3.* In addition, given the strong image generation capability of MLLM, we additionally compare our method against MLLM-based image generation under these conditions.

**Evaluation Metrics.** Our task is to generate photorealistic object images under topology and spatial layout constraints. Accordingly, we need to evaluate our results from two perspectives: image realism and structural fidelity. For image realism, we use the CLIP-based FID, where lower scores indicate greater similarity to real images. For structural fidelity, however, no existing metric can

---

[1] As recent ControlNet++ and ControlAR lacks scribble support, we select ControlNet and T2I-Adapter.

directly extract and compare topology and spatial layout information from images, making quantitative evaluation particularly challenging. The commonly used evaluation metrics such as SSIM or LPIPS are not designed to evaluate the alignment of topology and spatial layout, but rather focus on pixel-level or feature-level alignment. Therefore, we do not employ any objective metric to assess structural alignment, *and a detailed discussion on this choice is provided in Appendix A.3.4.* Instead, we conduct a Mean Opinion Score (MOS) study (Huynh-Thu et al., 2011), where human participants evaluate the alignment between the generated image and the input structure (**MOS-A**) on a 1–5 Likert scale (Likert, 1932). To further evaluate image realism, we also conduct this MOS study on the realism of generated images (**MOS-R**). *The specific form and scoring criteria of the human evaluation are provided in Appendix A.3.5.*

**Choice of MLLMs.** Our image generation algorithm is compatible with various MLLMs, such as GPT-4o (OpenAI, 2024), Gemini-2.5 (Team, 2025a), and Seedream-4 (Gao et al., 2025), as the underlying model for each component, and we therefore conducted experiments to assess the performance of different MLLMs in our task. Among them, GPT-4o can serve as the underlying model for all components in our framework, and it also achieved the best performance in our experiments(*detailed comparisons provided in Appendix A.5*). Moreover, in practice, GPT-4o's contextual memory enables seamless data transfer across different components of our algorithm, facilitating a more streamlined and efficient implementation of the overall pipeline. Therefore, all experiments presented in this section are conducted using GPT-4o as the underlying model.

### 4.2 Results & Analysis

#### 4.2.1 Comparison with Other Baselines

We compare the performance of our approach against other baseline methods. Related results are shown in Tab. 1-Left and Fig. 5.

**Realism of the Generated Images.** As shown in Tab. 1-Left, our method achieves the best performance on both FID scores and MOS-R ratings. We attribute this superiority to two key factors: (1) StructMap provides clear and straightforward structural guidance, and (2) our image generation algorithm effectively generate realistic details while faithfully preserving the specified structure. In contrast, other methods achieve worse performance on both FID scores and MOS-R ratings. Textual prompts often suffer from ambiguity or under-specification, while method based on lineart and scribble tend to over-emphasize edge alignment, sometimes at the cost of realism and natural visual appearance. Further, as shown in Fig. 5, our method generates object images with significantly better visual realism. The generated objects appear indistinguishable from real-world objects at first glance. In contrast, other methods may produce visual flaws, such as unnatural textures (Lineart & Controlnet++ result in example (a)) or distorted shapes (Scribble & Controlnet result in example (b)), which make the generated images appear synthetic. Additionally, when we replaced the image generation model with GPT-4o, the realism of the results improved; however, some irregular geometric shapes still appeared (Text & GPT-4o in example (b)), making the generated objects look less realistic. This could be due to the inability of text to fully describe the shape of objects.

**Structural Alignment with the Conditioning Input.** From Fig. 5, we observe that text and scribble conditioned generation yields weak structural alignment. This indicates that such conditions are less effective to convey precise structural constraints to the model, making it difficult to enforce accurate object structure. The image generated by Lineart & Controlnet++ strongly preserves the structural constraints encoded in the condition. However, this strength can be a double-edged sword: image generation models may overfit to low-level features(Lineart & Controlnet++ result in example (b)), reproducing input noise and prioritizing pixel consistency at the expense of realism. In comparison, our method consistently preserves the underlying topology and spatial layout. This is further supported by MOS-A scores in Tab. 1-Left, where our results perform much better than text and scribble and comparably to lineart, indicating that the structural fidelity is effectively preserved in human visual perception. Additionally, when we replaced the image generation model with GPT-4o, structural errors occasionally persisted (Text & GPT-4o in example (f) and Scribble & GPT-4o in example (e)) due to missing information in the text and scribble inputs. And when using lineart as input condition, MLLM yields visually comparable results against Struct2Real with noticeable decline of performance on metrics like MOS-R and MOS-A.

Figure 5: Qualitative results across different baseline methods. For each sub-panel, the top row shows the input conditions for each method, and the bottom two rows show the object images generated by different method. The icon in the bottom-right corner of each image indicates the method used to generate it, and the meaning of each icon is explained in the legend at the bottom. *Due to space limitation, more results and full textual descriptions are provided in Appendix A.4.* The text condition in the figure appears small and may require zooming in for a clear view.

### 4.2.2 ACCESSIBILITY OF DIFFERENT CONDITIONS

In addition to the quality of generated images, we additionally analyze the accessibility of each condition using creation time and creation difficulty metrics collected during dataset construction. Specifically, we 1) record the average time consumed to create each type of condition during the creation process, and 2) ask each creator to rate the difficulty of producing data of each type on a 1–5 Likert scale (Likert, 1932) based on their perceived mental effort and the level of technical or artistic skill required. As indicated by the results in Tab. 1-Right, lineart, despite its proficiency in guiding image generation, is considered the most complicated to produce among the four types. This significantly adds to the difficulties in the acquisition of high quality generated images. In comparison, StructMap is rated with a much less creation time and difficulty, making it the optimal choice of conditioning in controllable image generation.

### 4.3 ABLATION STUDY

To validate the design of Struct2Real, we conduct a series of ablation studies. Specifically, we first assess the generality of our image generation algorithm by replacing the underlying model in our image generation algorithm with other MLLMs. Second, we evaluate the effectiveness of StructMap by applying our image generation algorithm under alternative input conditions, including text, lineart, and scribble. Some experimental results are presented in Fig. 6. As shown in Fig. 6-top, our algorithm achieves consistently strong results when applied with all underlying MLLMs. In Fig. 6-bottom, we observe that although other conditioning inputs exhibit a substantial improvement in visual realism after employing our generation algorithm, they still suffer from considerable issues in structural fidelity (Text condition in example (a) and Scribble condition in example (c)). While images generated using lineart as condition attain performance comparable to those conditioned on StructMap in structural fidelity, their texture exhibits lower visual realism (Lineart condition in example (a) and example (c)), and the acquisition cost of lineart is much higher than that of StructMap. Furthermore, we validate the effectiveness of our image generation algorithm by comparing the results generated by our algorithm to those generated by directly employing an MLLM (both conditioned on StructMap). *Detailed results and analysis are provided in Appendix A.5, and*

Figure 6: **[Top]** Ablation study results across different underlying MLLMs. **[Bottom]** Ablation study results across different input conditions. For each sub-panel, the top row shows different types of structural conditions, and the bottom row shows the corresponding generated object images. The text condition in the figure appears small and may require zooming in for a clear view.

*we provide an additional experiment in Appendix A.11 that verifies the contribution of each module in the algorithm.*

### 4.4 DISCUSSION

#### 4.4.1 COMPARISON WITH 3D GENERATION METHODS

We observed that, beyond conventional image generation methods, one can also leverage 3D generation methods by first creating 3D objects and then rendering them into images. Therefore, we included several representative 3D generation methods for comparison. The results demonstrate that these methods perform poorly on the task of structure-grounded object image generation. *Detailed experimental results and analysis are provided in Appendix A.6.*

#### 4.4.2 GENERATING MULTI-VIEW IMAGES WITH STRUCTMAP

Our method supports generating consistent multi-view images of an object. Specifically, once a StructMap for an object is created, we can render StructMap images from different viewpoints and use each of them as a condition to generate multi-view images of the object with our image generation algorithm. *Detailed examples are provided in Appendix A.15.*

#### 4.4.3 GENERATING OTHER CATEGORIES OF IMAGES

To further showcase the generalization of our method, we conducted additional generation experiments, including: 1) generating articulated objects, 2) generating non-rigid objects, and 3) generating multi-object scenes, which demonstrates the strong generalization capability of our method. *Detailed methods and examples are provided in Appendix A.17.*

## 5 CONCLUSION

We propose Struct2Real, a novel framework for photorealistic object image generation under precise structural control. Central to our method is StructMap, an explicit 3D representation that encodes object topology and spatial layout via geometric primitives. Around StructMap, we design a structure modeling system, enabling users to conveniently design StructMaps from scratch. Then, we propose an image generation algorithm that works together with MLLM to generate visually realistic images that faithfully preserve both the topology and spatial layout of the input structure. Comprehensive experiments show that Struct2Real achieves high structural fidelity and strong visual quality with relatively low user effort. Struct2Real offers a new perspective on controllable image generation by introducing symbolic structural design. We hope this work may inspire the community at the intersection of structural representation and generative modeling.

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

# A APPENDIX

## CONTENTS

## A.1 Discussions about StructMap

**StructMap vs. Sketch** While both StructMap and sketches are able to describe an object's structure, they differ in dimensionality and expressiveness, which leads to different performance in structure-grounded object image generation. (1) Sketches are inherently 2D and manually drawn, requiring high professional skills yet still being difficult to faithfully capture 3D structural properties such as depth, angle of view, and symmetry. In contrast, StructMap is natively 3D, and thus naturally encodes such 3D attributes. Moreover, sketches struggle to maintain multi-view consistency, that is, creating coherent sketches of the same object from different viewpoints is nontrivial, as each new view requires redrawing and careful recalibration. In comparison, once a StructMap is constructed, rendering from different viewpoints becomes trivial and consistent, without requiring additional effort. (2) StructMap offers a more accessible structural input condition. Unlike sketches that demand drawing skills, StructMap creation only requires assembling geometric primitives via interpretable parameters. This significantly reduces the creation burden and skill barrier for users. Alternatively, attempts to simplify sketching, like using scribbles, often result in significant information loss, leading to incomplete or ambiguous structural specification.

**StructMap vs. Text, Semantic Layouts and Pose Keypoints** Compared to StructMap, coarse-grained conditions such as text, semantic layouts, and pose keypoints provide only coarse-grained descriptions of an object's structure. Specifically, textual descriptions are clearly insufficient for precisely conveying the structure of a 3D object, especially regarding the size, spatial layout and the connections between different components. Semantic layouts can only specify the semantic category of each spatial region, but they are still unable to precisely capture the topology and shape of an object, and their characterization of the spatial layout is also insufficiently detailed. Finally, pose keypoints only provide information about topological connectivity and cannot capture the shape or size of each component. We provide an example in Fig. 7 to illustrate these points more intuitively.

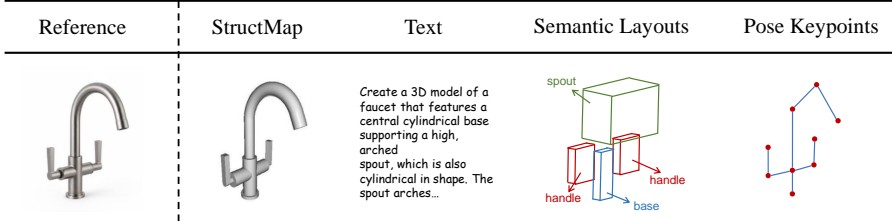

| Reference | StructMap | Text | Semantic Layouts | Pose Keypoints |
|---|---|---|---|---|

Figure 7: An example of how StructMap, text, semantic layouts, and pose keypoints express object structure.

**Structure Modeling System vs. CAD** Unlike traditional Computer-Aided Design (CAD) tools, which are primarily geared towards precise and detailed 3D modeling typically used in engineering and industrial design, our structure modeling system focuses on simplicity, accessibility, and task-specific functionality. CAD systems often require significant professional expertise, extensive training, and a deep understanding of geometrical constraints(e.g., parallelism, symmetry, perpendicularity, concentricity). They are powerful, but their complexity can be a barrier for non-expert users or for applications where such precision is unnecessary. In contrast, our structure modeling system is designed specifically for structural representation rather than comprehensive 3D object modeling. It provides an intuitive, streamlined interface that allows users—even those without a background in engineering or design—to quickly create, manipulate, and visualize structural models. By prioritizing usability and focusing on the essential features needed for structure modeling, our system significantly lowers the entry barrier, enabling easy use by anyone.

## A.2 Higher-order Configurations in StructMap Design Interface

In addition to primitive-level composition, our design interface supports a collection of higher-order configurations distilled from frequently occurring structural patterns in real-world objects. We impose certain constraints on the degrees of freedom of the primitives within these higher-order configurations and design corresponding parameters to control the adjustable attributes of the primitives in these configurations. This allows users to instantiate a higher-order configuration to create multiple primitives simultaneously, enabling more primitives to be created with fewer parameters and thereby

improving creation efficiency. For example, in symmetric structures, we can use a single parameter to jointly control the parameters of the two primitives on both sides of the symmetry axis. We have designed a total of 25 higher-order configurations, and representative examples of the higher-order configurations are visualized in Figure 8.

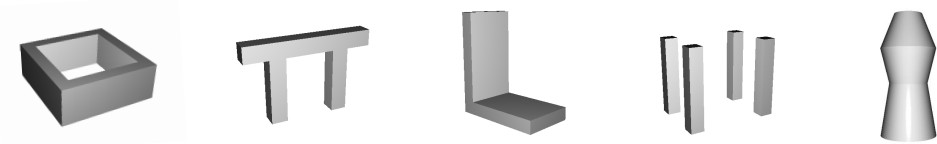

Figure 8: Representative examples of the higher-order configurations.

### A.3 DETAILED EXPERIMENT SETTINGS

#### A.3.1 DATASET CONSTRUCTION

To construct our structure-prior dataset, we first invite human creators to build a diverse set of StructMaps that span a wide range of object categories, topologies, and spatial layouts using our StructMap design system. Based on each designed StructMap, the creators are then instructed to manually create corresponding textual descriptions, linearts, and scribbles that depict the same underlying structural configuration. When drawing linearts and scribbles, the creators are free to choose a suitable viewpoint that they feel best conveys the structure. This ensures that each set of inputs — text, lineart, scribble, and StructMap — corresponds to the same structural intent.

For the linearts, creators are encouraged to preserve as much structural detail as possible, closely match the reference StructMap, and apply light shading where appropriate to enhance three-dimensionality. However, in practice, we found that most creators, due to varying levels of technical skill, often produce line drawings with errors and noise. Since our dataset aims to provide high-quality lineart, we leverage GPT-based optimization techniques to enhance the line drawings, thereby reducing the technical demands and time investment required from creators, while still ensuring that the resulting lineart meets the demands of our dataset. Scribbles, in contrast, are designed to emphasize simplicity and accessibility — creators only need to roughly retain the main structure of the object, and we do not impose strict quality requirements.

It is important to note that our dataset construction does not require specific consideration of artistic style, as our primary focus is on the structural information within the data. Artistic style often influences the fine-grained appearance details of the objects, which are not the main concern of this dataset.

To help readers better understand the acquisition process and difficulty level of each type of input condition, we recorded a video showcasing the manual steps involved in creating all four types of inputs side by side. This can be found in the supplementary video file `all_conditions_creation.mp4`.

#### A.3.2 IMPLEMENTATION DETAILS

We use `Open3D` to render the 3D StructMap into 2D images with default settings, including Lambertian shading, neutral directional lighting, and uniform gray mesh color. The rendered images have a resolution of $512 \times 512$ pixels and are captured from carefully selected viewpoints to ensure key structural features are clearly visible. The image generation algorithm based on GPT-4o is executed via the official GPT-4o API. Full prompts used in our image generation algorithm are provided in Section A.7.

#### A.3.3 BASELINE METHODS

To comprehensively evaluate our method, we compare against strong baselines under each conditioning input. For each condition, we select two representative models that are either widely adopted

or represent state-of-the-art advancements in that condition. Here, we will provide a detailed description of each method we have selected.

- **Text-based Condition.** We compare against **OmniGen**(Xiao et al., 2024), a unified image generation model that excels in multiple domains. OmniGen demonstrates competitive text-to-image generation capabilities and inherently supports a variety of downstream tasks, such as controllable image generation and classic computer vision tasks. We also compare against **Lumina-Image 2.0**(Qin et al., 2025), a unified and efficient T2I generative framework. Lumina-Image 2.0 adopts a unified architecture (Unified Next-DiT) that treats text and image tokens as a joint sequence, allowing for seamless cross-modal interaction and enabling task expansion without retraining, leading to improved image fidelity and alignment with text inputs.

- **Lineart-based Condition.** We compare against **ControlNet++**(Li et al., 2024), an enhanced version of ControlNet that improves controllability by leveraging efficient consistency feedback mechanisms. Additionally, we evaluate **ControlAR**(Li et al., 2025), an advanced framework that integrates spatial controls into autoregressive image generation models. It generates the next image token based on control and image tokens' fusion, similar to positional encoding, thus enhancing the autoregressive model's control capabilities.

- **Scribble-based Condition.** We compare against **ControlNet**(Zhang et al., 2023), a widely used framework for guided image generation from edge and sketch inputs. ControlNet enables the generation of high-quality images based on simple sketches or outlines, allowing for precise control over the structure and composition of the generated visuals. Another comparison is with **T2I-Adapter**(Mou et al., 2023), a lightweight plug-and-play adapter that facilitates structure guidance for existing diffusion models. T2I-Adapter achieves structural control over the generation process by learning the alignment between internal knowledge of text-to-image models and external control signals.

### A.3.4 DISCUSSION ON EVALUATION METRICS

In our work, we did not use common evaluation metrics such as SSIM (Structural Similarity Index) and LPIPS (Learned Perceptual Image Patch Similarity), which are often used in other works for evaluating structural alignment. The primary reason for this is that these metrics are not designed to evaluate the alignment of topology and spatial layout, but rather focus on pixel-level or feature-level alignment. Our work places a significant emphasis on the alignment of topology and spatial layout, which go beyond pixel or feature accuracy and involve a more abstract understanding of topological configurations and spatial organization. Currently, there is no metric capable of precisely evaluating the alignment of topology and spatial layout, so we rely on human evaluation. This allows for a more nuanced evaluation of the generated images based on topology and spatial layout, which cannot be adequately measured by metrics like SSIM and LPIPS.

### A.3.5 HUMAN EVALUATION

We conduct a two-part Mean Opinion Score (MOS) study (Huynh-Thu et al., 2011), where human participants evaluate (1) the realism of generated images (**MOS-R**) and (2) the alignment between the generated image and the input structure (**MOS-A**) on a 1–5 Likert scale (Likert, 1932). Fig. 9 shows an example of our scoring system.

The detailed scoring criteria are defined as follows:

**MOS-R (Realism):**

- **1 – Very Poor:** The image is clearly artificial and unrealistic.
- **2 – Poor:** The image contains many unrealistic artifacts and lacks visual plausibility.
- **3 – Fair:** The image is somewhat realistic but contains noticeable flaws or inconsistencies.
- **4 – Good:** The image appears mostly realistic with only minor issues.
- **5 – Excellent:** The image is highly realistic and could plausibly be mistaken for a real photograph.

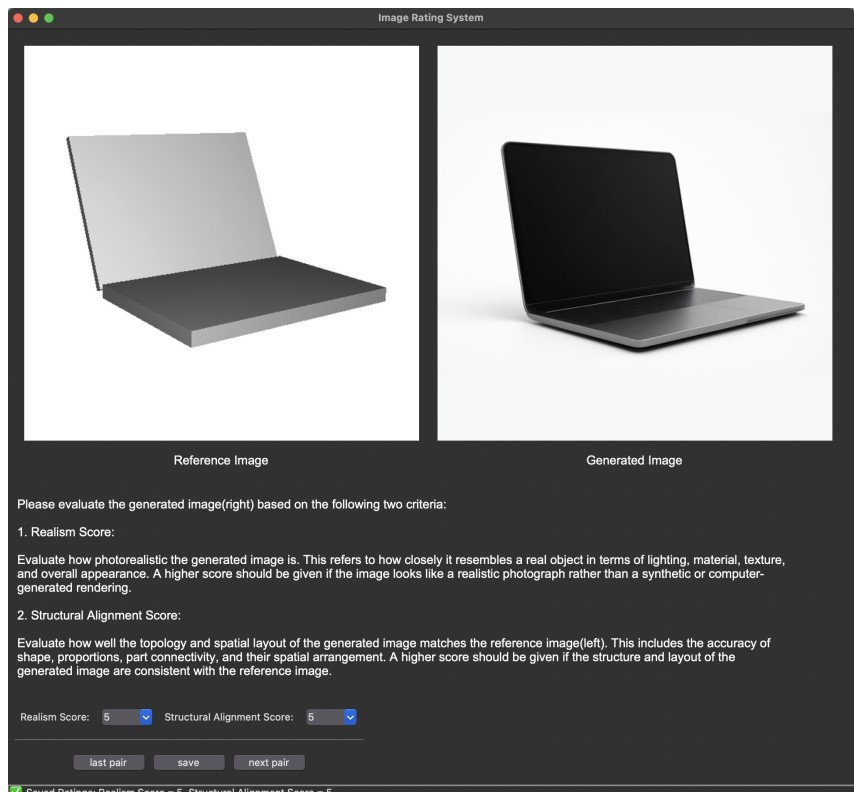

Figure 9: An example of the scoring system.

**MOS-A (Alignment with Structure):**

- **1 – Very Poor:** The image does not reflect the input structure at all.
- **2 – Poor:** Only a few structural elements are preserved; major components are missing or distorted.
- **3 – Fair:** The overall layout is partially aligned with the input, but several elements are misrepresented.
- **4 – Good:** The image largely follows the structural configuration, with only minor deviations.
- **5 – Excellent:** The image shows an almost perfect correspondence with the input structure.

A.4    MORE QUALITATIVE RESULTS

To further demonstrate the generality and effectiveness of Struct2Real across diverse object categories, we provide additional qualitative comparisons in Fig. 10 and Fig. 11, and further analysis the performance based on these results.

**Realism of the Generated Images.** Our qualitative results further confirms that our method consistently generates object images with higher visual realism. In many cases, the generated results appear convincingly realistic, with photorealistic textures, natural shading, and plausible materials. By contrast, baseline methods often display visual artifacts such as distorted geometry(Scribble & Controlnet in example (c)) or artificial textures(Lineart & Controlnet++ in example (g)).

**Structural Alignment with the Conditioning Input.** From visual comparisons, we find that text- and scribble-conditioned outputs often exhibit incorrect structural attributes, such as misplaced components(Scribble & Controlnet in example (i)) or missing parts(Text & Omnigen in example (f)), due to the ambiguity or incompleteness of the input. Lineart provides much stronger constraints, but may inadvertently introduce noise(Lineart & Controlnet++ in example (e)) or overfitting to edges(Lineart

& controlar in example (h)), which can hinder natural appearance. In contrast, our method reliably maintains key structural properties, even if minor variations in orientation or scale occasionally occur.

Additionally, due to space limitations in the figure, we do not show the full text conditions. Below, we provide the complete text conditions for (a)–(d) as illustrative examples.

```
(a) Create an image of an object structured as
follows:  The object is a pot with a conical body
tapering from a wider top to a narrower base.  It
has a circular lid that fits tightly on top, which
also carries a small cylindrical handle or knob at its
center.  There are two rectangular handles on either
side of the pot situated at the midpoint of the body.
The overall geometric form is symmetrically balanced
around the vertical axis, with the lid protruding
slightly beyond the top edge of the pot's body.

(b) Generate an 3D model of a laptop.  The laptop
should consist of two flat rectangular planes
connected along one edge, forming an angled,
hinge-like configuration.  The bottom plane lies flat
to represent the base, while the top plane is upright
at an angle to represent the screen.  Ensure the
proportions resemble a typical laptop design, where
the screen is smaller than or equal to the base in
size.

(c) Create a 3D model of a mug.  The mug should have
a cylindrical body that is open at the top to form a
hollow container.  It should also include a single,
circular handle that is attached perpendicularly to
the side of the cylinder.  The handle should form
a closed loop, resembling a torus, and should be
proportional in size to allow for practical gripping.
Ensure that the overall structure is smooth and
geometrically cohesive, with the hollow space inside
the cylinder and the handle forming two distinct yet
connected components.

(d) Create a 3D model of a pen.  The object is
cylindrical and elongated with a tapering tip at
one end that narrows into a pointed structure.  The
opposite end is capped with a flat, circular top.
Midway along the cylindrical body, a thin, rectangular
clip structure is attached, extending outward parallel
to the length of the cylinder.  Ensure that the
geometric composition includes symmetrical proportions
and smooth transitions between the tapered and
cylindrical sections.
```

## A.5 ABLATION STUDIES

To further analyze the effectiveness of Struct2Real in generating photorealistic object images under topology and spatial layout constraints, we conduct a series of ablation studies. These studies are designed to address the following questions: 1) How generalizable is our image generation algorithm across different underlying MLLMs? 2) What is the contribution of StructMap compared to other condition types (e.g., text, scribble and lineart)? 3) How effective is our proposed image generation algorithm?

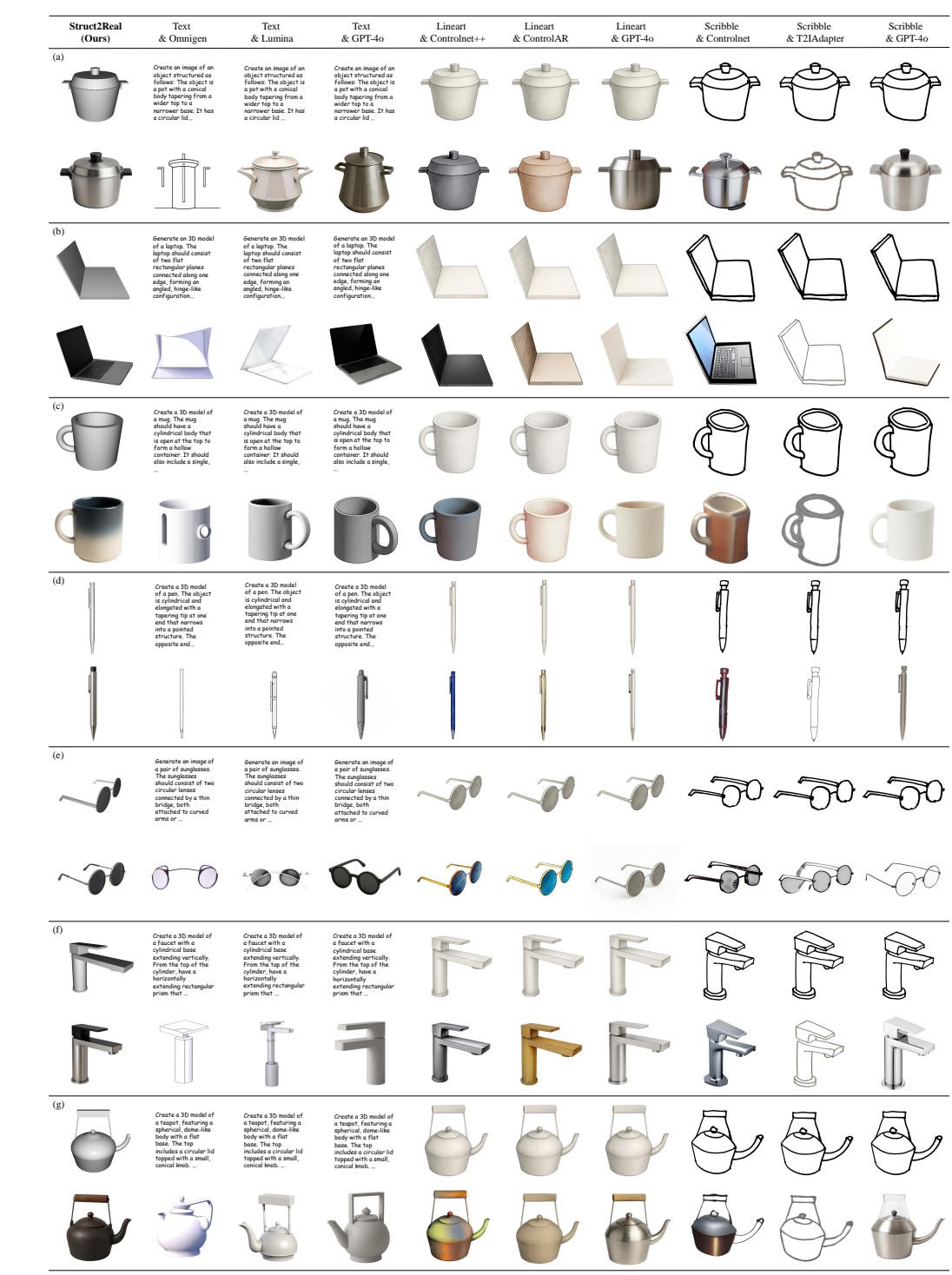

Figure 10: More qualitative results across different structural conditions and approaches. There are example (a)-(g). For each sub-panel, the top row shows different types of structural conditions, and the bottom row shows the corresponding generated object images. The text condition in the figure appears small and may require zooming in for a clear view.

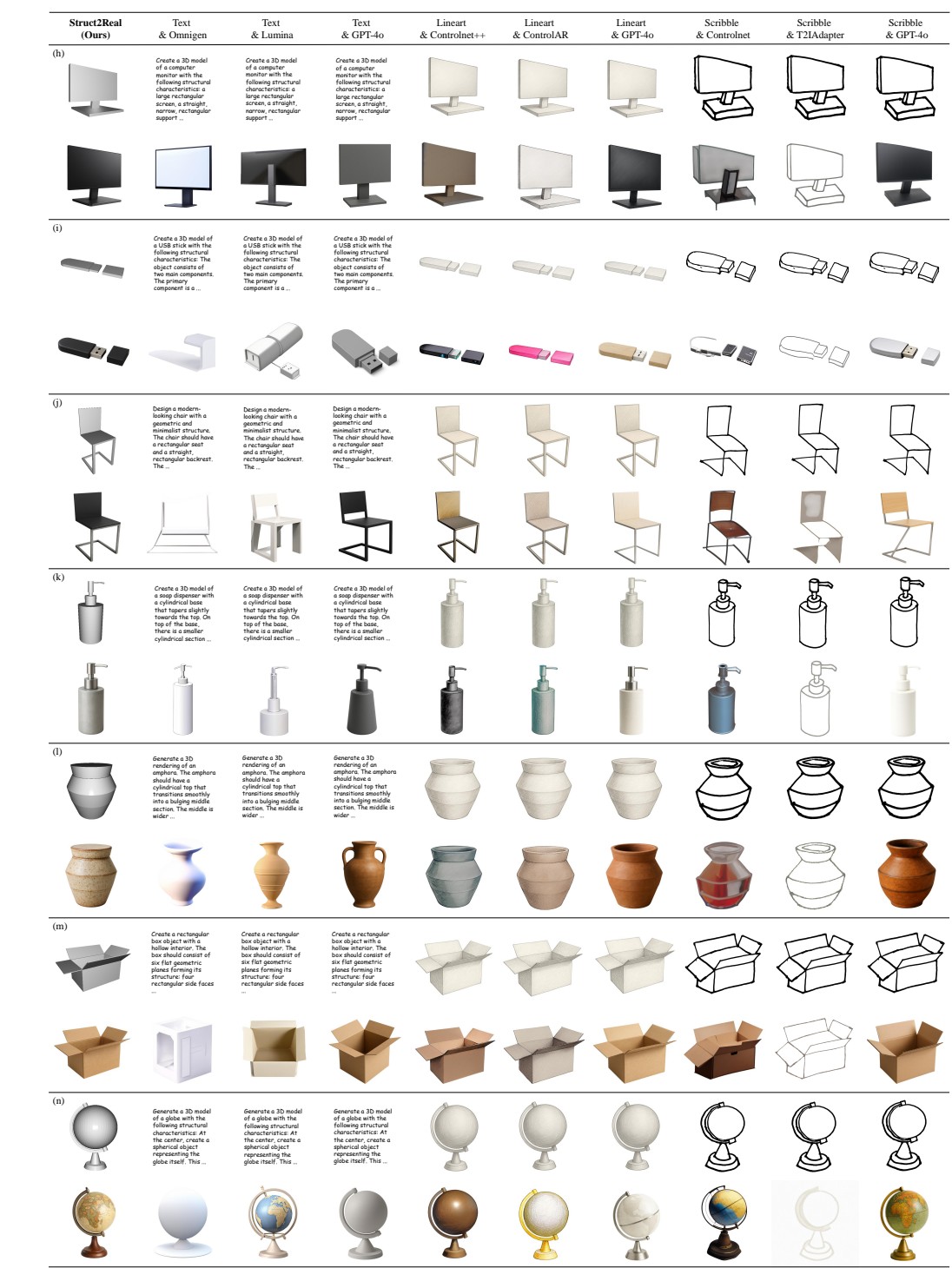

Figure 11: More qualitative results across different structural conditions and approaches. There are example (h)-(n). For each sub-panel, the top row shows different types of structural conditions, and the bottom row shows the corresponding generated object images. The text condition in the figure appears small and may require zooming in for a clear view.

### A.5.1 GENERALITY OF THE IMAGE GENERATION ALGORITHM UNDER DIFFERENT MLLMS

To assess the generality of our image generation algorithm, we replace the underlying MLLM of the three components with alternative models. Specifically, we perform two sets of experiments: i) substituting the MLLM in the image generator with four alternative models — Gemini-2.5 (Team, 2025a), Qwen-image-edit (Wu et al., 2025), Flux-1-kontext-dev (Labs et al., 2025), and Seedream-4 (Gao et al., 2025), and ii) substituting the MLLM in the condition augmentation module and the structural consistency discriminator with another model — Gemini-2.5 (Team, 2025a). For each experiment, we first compare the performance of our algorithm under different MLLMs against baseline methods, and then conduct comparisons across different MLLMs. Related results are presented in Fig. 12 and Fig. 13.

From Fig. 12, we observe that baseline methods (right) often produce results with limited visual realism (e.g., Text & Lumina result in example (c)) or noticeable structural inconsistencies (e.g., Scribble and Controlnet result in example (e)). In contrast, our algorithm consistently yields more realistic and structurally faithful results across all tested MLLMs. Among them, results generated using GPT-4o as the underlying model exhibit the highest visual realism (e.g., StructMap & GPT-4o result in example (f)). From Fig. 13, we observe that both underlying MLLMs consistently outperform baseline methods. Moreover, results generated with Gemini-2.5 exhibit visual realism and structural consistency comparable to those obtained using GPT-4o (e.g. StructMap & Gemini-2.5 in example (e)), which further confirms the generality of our algorithm across different MLLMs.

### A.5.2 THE EFFECTIVENESS OF STRUCTMAP

To evaluate the effectiveness of StructMap, we replace the input to our image generation algorithm with alternative conditions – text, lineart, and scribble. These conditioning inputs are obtained from our structure-prior dataset, where all conditions are aligned to the same object structure. To ensure fairness and correctness, we slightly adjust the prompting strategy of our algorithm for different input types. We compare the performance of StructMap with the three alternative conditions. Related results are shown in Fig. 14.

From Fig. 14, we observe that while all four conditioning inputs produce outputs of similar visual realism, their structural consistency varies significantly. Specifically, results conditioned on text and scribble exhibit poor alignment with the intended structures (e.g. Text result in example (c) and Scribble result in example (d)). This indicates that such conditions provide insufficient structural information, limiting the algorithm's ability to generate structurally faithful images. In contrast, conditioning on our proposed StructMap produces outputs that reliably preserve the underlying topology and spatial layout, highlighting its effectiveness in providing accurate structural guidance. While results conditioned on lineart achieve a comparable level of structural consistency, StructMap offers clear advantages in terms of acquisition efficiency and ease of creation, making it a more practical and scalable option for structure-grounded generation.

### A.5.3 THE EFFECTIVENESS OF IMAGE GENERATION ALGORITHM

To evaluate the effectiveness of our proposed image generation algorithm, we compare its outputs with those generated by directly employing a single MLLM (both conditioned on StructMap). Qualitative results are presented in Fig. 15. As shown, results obtained by directly using an MLLM often suffer from structural inconsistencies (e.g. w/o algorithm result in example (a) and (b)) or reduced visual realism (e.g. w/o algorithm result in example (d) and (f)). In contrast, our algorithm consistently produces images that are both visually realistic and structurally faithful, thereby validating its effectiveness.

To further investigate the contribution of each component in our algorithm, we evaluate four component configurations: (i) image generator only, (ii) combining the image generator with the condition augmentation module, (iii) combining the image generator with the structural consistency discriminator, and (iv) the full algorithm with all three components. An illustrative example is presented in Fig. 16. From Fig. 16, we observe that the condition augmentation module enhances visual realism of outputs, such as producing smoother transitions between different object parts (example (ii) and (iv)). Without the structural consistency discriminator, outputs often display structural mismatches with the input, such as incorrectly shaped faucet knobs (example (i)), whereas including it yields

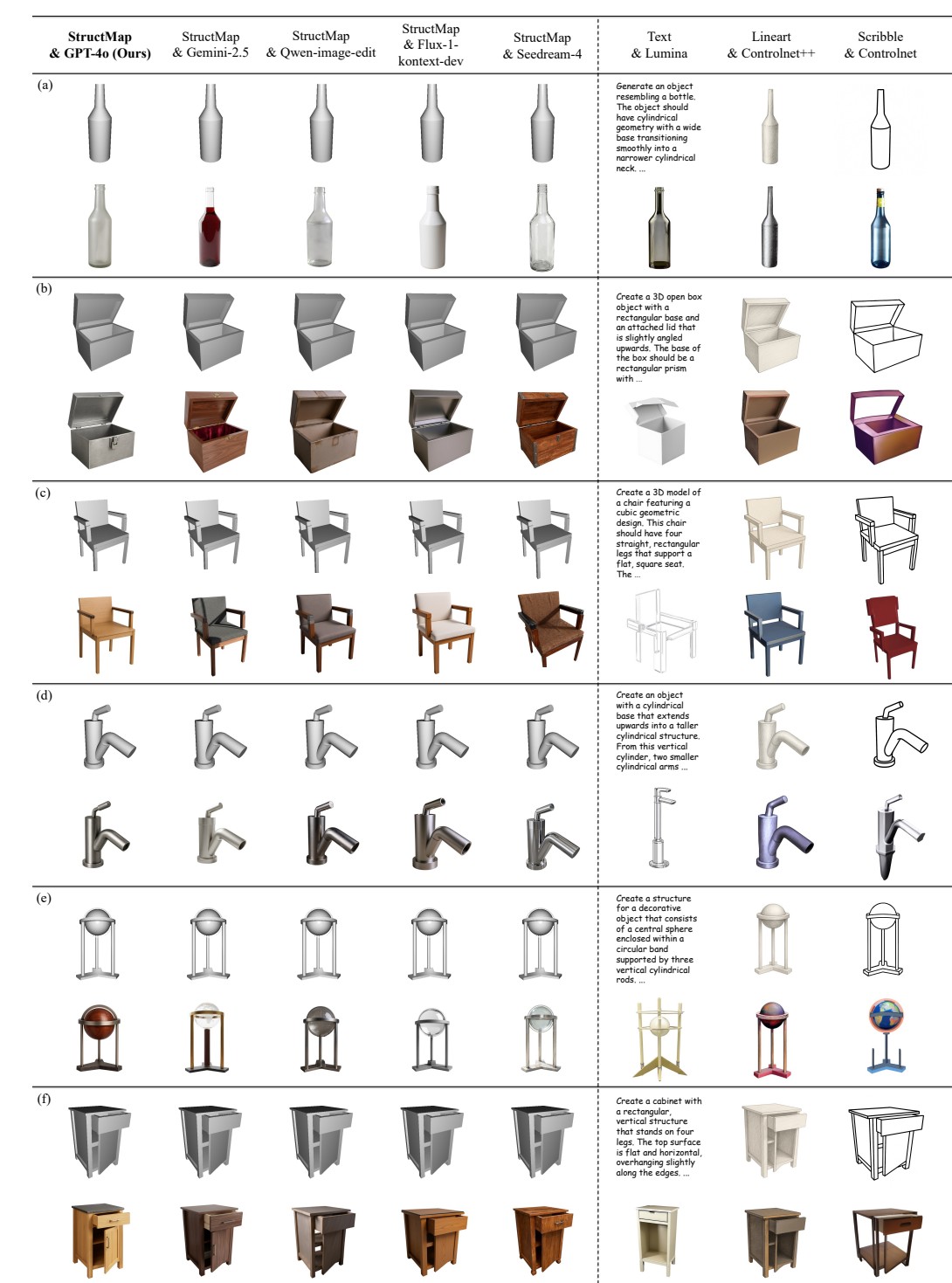

Figure 12: Qualitative comparison of generated images when the underlying model of the image generator is substituted with four alternative MLLMs. Generated outputs are compared against baseline methods. For each sub-panel, the top row shows the input structural conditions, while the bottom row shows the corresponding generated object images. The text condition in the figure appears small and may require zooming in for a clear view.

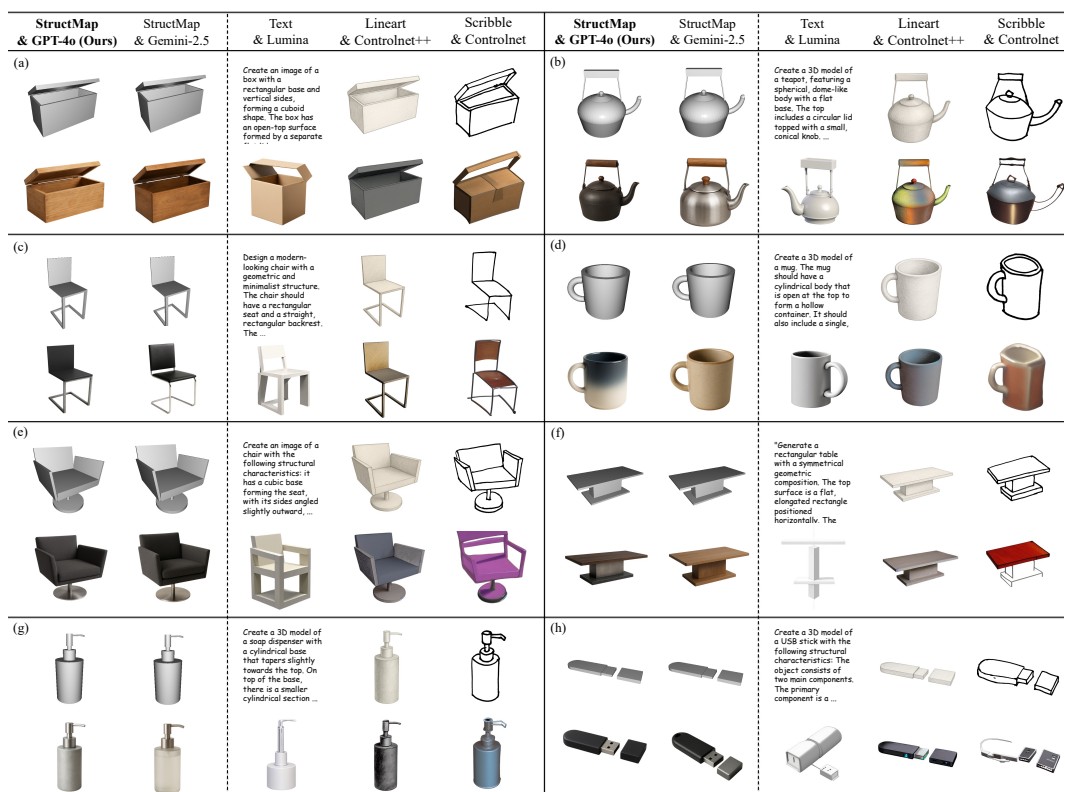

Figure 13: Qualitative comparison of generated images when the MLLM backbone of the condition augmentation module and the structural consistency discriminator is substituted with Gemini-2.5-flash. Generated outputs are compared against baseline methods. For each sub-panel, the top row presents the input structural conditions, while the bottom row displays the corresponding generated object images. The text condition in the figure appears small and may require zooming in for a clear view.

more structurally faithful results (example (iii) and (iv)). These findings demonstrate that each component plays an essential role in achieving high-quality, structure-grounded image generation.

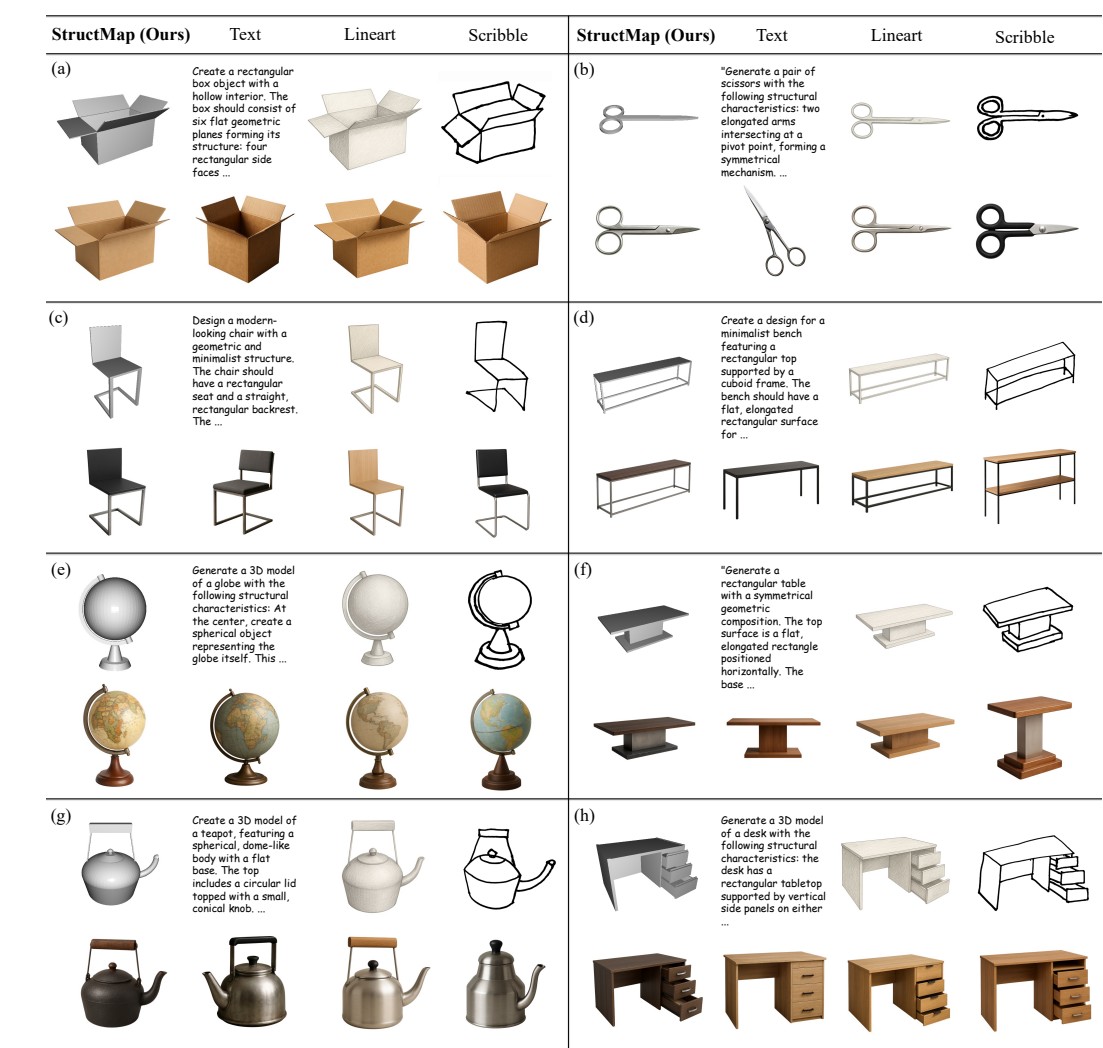

Figure 14: Qualitative comparison of image generation results under different conditioning inputs. For each sub-panel, the top row shows different types of structural conditions, and the bottom row shows the corresponding generated object images. The text condition in the figure appears small and may require zooming in for a clear view.

## A.6 COMPARISON WITH 3D GENERATION METHODS

We observed that, beyond conventional image generation methods, one can also leverage 3D generation methods by first creating 3D objects and then rendering them into images. So in this subsection, we evaluate the performance of 3D generation methods on our proposed task of structure-aware image generation. Specifically, we use four types of conditions from our structure-prior dataset, apply state-of-the-art text-to-3D or image-to-3D methods to generate 3D meshes, and then render images from these meshes to obtain realistic object images. In practice, we employ Hunyuan3D (Team, 2025b)) as the text-to-3D and image-to-3D model in our experiments.

As shown in Fig. 17, the performance of these 3D generation methods is generally unsatisfactory. When using text and scribble as conditions, the results exhibit very poor structural fidelity, since text and scribble cannot accurately and completely convey structural information. By contrast, when conditioned on lineart and StructMap, the generated results achieve better structural fidelity but suffer from low realism. This is because image-to-3D methods are designed to faithfully reconstruct the objects present in the input images. Consequently, imperfections in lineart drawings and discontinuous connections between geometry primitives in StructMap are also preserved. Moreover, the

| | StructMap (Input) | **Struct2Real (Ours)** | w/o algorithm | | StructMap (Input) | **Struct2Real (Ours)** | w/o algorithm |
|---|---|---|---|---|---|---|---|
| (a) | | | | (b) | | | |
| (c) | | | | (d) | | | |
| (e) | | | | (f) | | | |

Figure 15: Qualitative comparison between our proposed image generation algorithm and directly employing a single MLLM, both conditioned on StructMap.

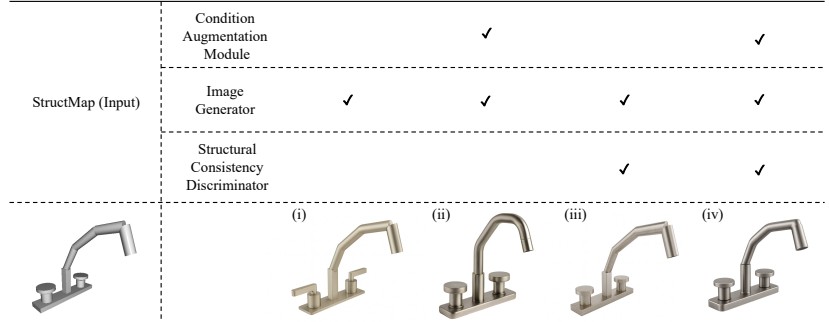

| | | | | | |
|---|---|---|---|---|---|
| | Condition Augmentation Module | | ✓ | | ✓ |
| StructMap (Input) | Image Generator | ✓ | ✓ | ✓ | ✓ |
| | Structural Consistency Discriminator | | | ✓ | ✓ |
| | | (i) | (ii) | (iii) | (iv) |

Figure 16: Illustrative example of the ablation study on different component configurations: (i) image generator only, (ii) image generator + condition augmentation module, (iii) image generator + structural consistency discriminator, and (iv) the full algorithm.

textures of the generated meshes are determined by those in the input images. As our conditional images do not provide any texture information, the resulting meshes inevitably contain unrealistic textures.

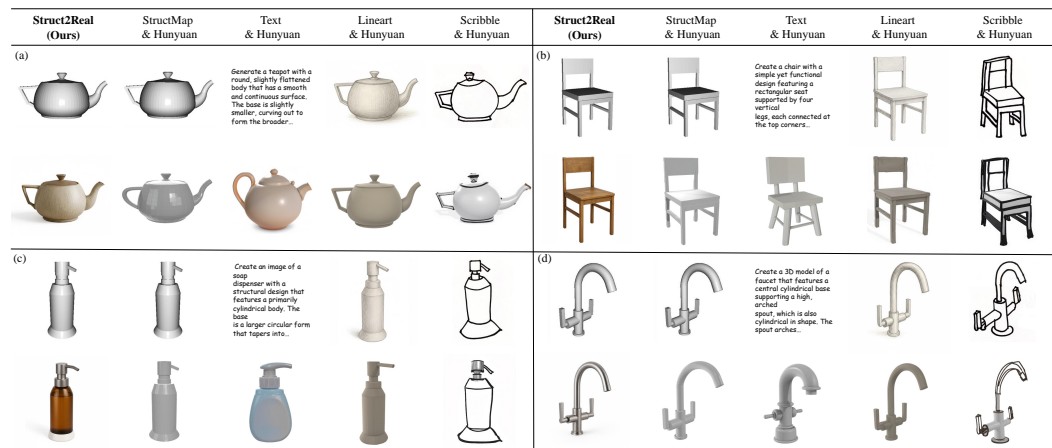

Figure 17: Qualitative results across different 3d generation methods. For each sub-panel, the top row shows the input conditions for each method, and the bottom row displays the corresponding generated object images. The text condition in the figure appears small and may require zooming in for a clear view.

## A.7 DETAILED PROMPTS AND RESPONSES IN IMAGE GENERATION ALGORITHM

### A.7.1 CONDITION AUGMENTATION MODULE

**StructMap Properties.** This prompt introduces the core properties of StructMap by clarifying that the StructMap only encodes the structural properties of an object while omitting surface-level appearance.

```
This image shows an object composed of basic geometric
primitives (such as cuboids, cylinders and spheres).
This model is not textured or shaded, but conveys
precise topology and spatial layout of the object.
```

**Instruction 1.** This prompt instruct the model to interpret the structural configuration by describing the StructMap's topology and spatial layout.

```
Please analyze and describe the object's 3D structural
features, focusing especially on:
- the number, types, and detailed geometric features
of base primitives used | for each distinct part,
describe its shape precisely (e.g., is the cylinder
tapered or uniform?  is the cube stretched or
flattened?  are any primitives angled, tilted,
or distorted?), along with its approximate size,
proportions, and individual orientation;
- the relative sizes and spatial relationships between
parts | describe which parts are significantly
larger/smaller, which ones enclose others or sit on
top, and how the components are scaled or arranged
in relation to each other; - how these parts are
connected (e.g., touching, intersecting, aligned);
- the viewing angle and orientation of the object
(e.g., is it viewed from front, side, top-down, or at
an isometric angle; and which direction is the object
facing).
```

**Structural Textual Description. (Model Response)** This is MLLM's response in the condition augmentation module.

Here is a detailed 3D structural analysis of the
chair-like object, including its geometric primitives,
spatial relationships, connections, and viewing
orientation:

1.  Primitive Shapes and Detailed Features The object
is composed entirely of cuboids (rectangular prisms).
Here's a breakdown:
a.  Backrest
Shape:  Vertically oriented cuboid
Size & Proportions:  Wide and flat (stretched along
the horizontal and vertical axes, thin along the
depth)
Orientation:  Upright; perpendicular to the seat
Position:  Attached to the rear of the seat, elevated
above it
b.  Seat
Shape:  Flat cuboid
Size & Proportions:  Square top surface, shallow
height
Color:  Darker than the rest of the structure
(possibly meant to indicate cushion material)
Orientation:  Horizontally flat, aligned with the base
plane
Position:  Central structural element, connects all
leg elements and backrest
c.  Legs (4)
Shape:  Uniform cuboids (not tapered)
Size & Proportions:  Tall and narrow (long vertical
axis, short cross-section)
Orientation:  Vertically aligned, each positioned near
a seat corner
Position:  Extend downward from the seat to the floor
d.  Support Bars (3 visible)
Shape:  Thin cuboids
Size & Proportions:  Thin and narrow; shorter than
legs
Orientation:  Horizontally aligned
Position:  Two connect the front and back legs (left
and right side), one connects the side legs at the
back
Function:  Likely structural reinforcement

2.  Relative Sizes and Spatial Relationships
Backrest is vertically the tallest single component
and thinner than the seat.
Seat is the largest surface area component,
proportionally wider than it is tall.
Legs are all of equal size and symmetrically placed
under the seat.
Support bars are thinner and shorter than legs but
positioned at mid-height between the seat and floor,
maintaining the structural frame.

3.  Connectivity and Alignment
Backrest is directly attached to the rear of the seat.
Legs are flush-mounted to the four corners of the seat
underside | no overhang.
Support bars are aligned with leg centers and

```
intersect or touch the inner faces of the legs.
All joints appear cleanly orthogonal | no visible
rotation, tapering, or angular misalignment.

4.  Viewing Angle and Orientation
The object is rendered from a slightly elevated
isometric view | a 3D angle showing:
Front-right perspective (both front and right legs are
visible)
Top surface of the seat is also partially visible
The backrest is facing away from the viewer
Camera is positioned at an angle slightly above and to
the right of the object's center, creating a 3-point
perspective-like view
```

### A.7.2   IMAGE GENERATOR

**Instruction 2.**   This prompt instructs the model to generate a photorealistic object image based on the StructMap and the structural description.

```
Please generate a realistic image of this object |
keeping the camera angle and perspective identical to
the original model I provided.
Your output should be a single-view image (like a
product photo or concept rendering) that:
- strictly preserves the structure, topology, spatial
layout, and proportions described in the previous
analysis,
- enhances visual realism through texture, material,
lighting, and shading, without altering any structure,
- maintains precise alignment with the viewing angle,
orientation, and relative scale described earlier.

Important:
- If this is the first generation attempt, simply
follow the instructions above.
- For subsequent attempts:  the previous generation
produced a failed and inconsistent result.  Please
refer to the reasons for inconsistency.  Carefully
review the prior output and correct the specific
issue(s) that caused the failure | do not repeat the
same mistake(s).  Use the last result as a guideline
for what to avoid.  Ensure the final render strictly
preserves the structure, topology, spatial layout,
proportions, and camera alignment described above,
and only enhances visual realism (textures, materials,
lighting, and shading) without altering any shapes or
connectivity.
```

**Style Prompt.**   To support further control, our algorithm also allows users to optionally provide a free-form prompt to specify visual style.

```
Style:  A rosewood/plastic chair.
```

### A.7.3   STRUCTURE CONSISTENCY DISCRIMINATOR

**Instruction 3.**   This prompt instructs the model to compare the generated image with the StructMap and the structural description, and provide a conclusion and a reasoning.

```
Please judge if the generated image is consistent
with the StructMap in geometric structure (or the
structural description you previously provided).
Focus specifically on topology, geometric shapes,
proportions, spatial arrangement – do not prioritize
differences in textures, colors, or materials.
And Provide a conclusion with reasoning.
```

**Consistency judgment and reasoning. (Model Response)**    This is MLLM's response in the structural consistency discriminator.

```
Conclusion:
The generated realistic image is highly consistent
with the original StructMap and the detailed
structural description I previously provided.
The geometric topology, proportions, and spatial
arrangement match precisely.

Reasoning:
1.  Topology & Primitives
The chair is built entirely from straight,
axis-aligned rectangular cuboids|seat slab, four
vertical legs, apron rails, and a single upper
backrest plank.
The generated image preserves all these elements:
no extra components, no curves or tapering, and no
missing members.
2.  Geometric Shapes & Proportions
Seat:  Remains a thin, flat rectangular slab with
roughly square plan.
Backrest:  A single horizontal plank aligned with the
rear legs and rising above the seat, leaving a clear
gap beneath|exactly as described.
Legs:  Four vertical posts of uniform thickness.
Rails/Stretchers:  Horizontal rectangular bars just
below the seat and lower down for bracing.
Relative thicknesses (legs slightly thicker than
rails, seat thicker than rails but thinner than legs)
match the earlier proportions.
3.  Spatial Arrangement & Alignment
Legs remain at the four corners, vertical and
perpendicular to the seat.
Rails and stretchers connect opposing legs cleanly,
square to the ground.
Backrest plank sits flush between the rear legs.
The camera perspective|three-quarter isometric
view from slightly above the front-left|matches the
original model's orientation.
```

A.8    THE USE OF LLM

This work employs large language models (LLMs) in two distinct ways. First, our proposed method incorporates multimodal large language models (MLLMs), and all implementation details of their usage are thoroughly described in the main text and the appendix. Second, during paper writing, we used an LLM to polish and refine certain passages for clarity and readability.

## A.9 GENERATION OF OBJECTS WITH MORE COMPLEX STRUCTURES

To further demonstrate the capability of our method, we selected several more complex object examples for additional visualization, as shown in Fig. 18. From Fig. 18, we can observe that our method also produces highly realistic and structurally consistent images for more complex examples, demonstrating its strong capability in handling complex objects. Moreover, in Fig. 19, we present a comparison between the StructMaps and their corresponding lineart, along with the time required to create each of them. As shown in Fig. 19, all StructMaps were created within 15 minutes, while creating the corresponding lineart typically took more than 30 minutes, demonstrating that our method achieves higher efficiency.

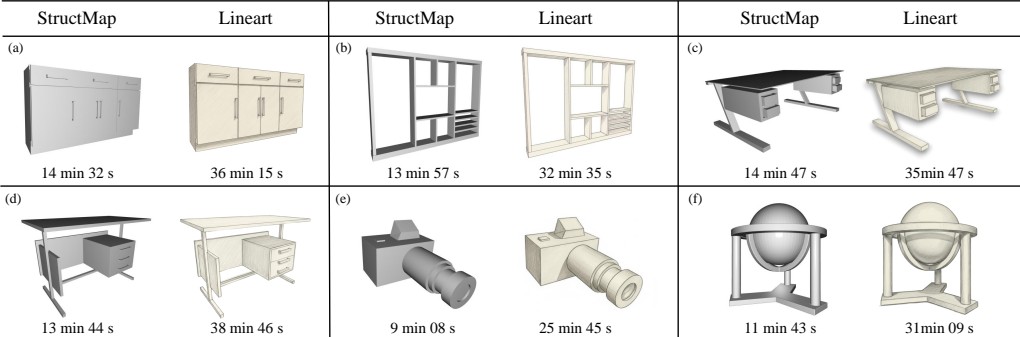

Figure 18: Examples of image generation for objects with more complex structures.

| StructMap | Lineart | StructMap | Lineart | StructMap | Lineart |
|---|---|---|---|---|---|
| (a) | | (b) | | (c) | |
| 14 min 32 s | 36 min 15 s | 13 min 57 s | 32 min 35 s | 14 min 47 s | 35min 47 s |
| (d) | | (e) | | (f) | |
| 13 min 44 s | 38 min 46 s | 9 min 08 s | 25 min 45 s | 11 min 43 s | 31min 09 s |

Figure 19: The StructMaps and their corresponding lineart for objects with more complex structures. The creation time for each condition is shown below it.

## A.10 THE COMPLEXITY OF STRUCTMAPS IN OUR DATASET

To more comprehensively illustrate the complexity of StructMaps in our dataset and the relationship between complexity and creation difficulty, we present in Fig. 20 the distribution of the number of primitives in our StructMaps, as well as the average creation time for StructMaps with different primitive counts. From Fig. 20, we can observe that our dataset contains StructMaps with a wide range of complexities, and most StructMaps consist of 7-13 primitives. And the creation time of a StructMap is approximately linearly correlated with the number of primitives it contains. Moreover, we provide several representative examples of StructMaps with different numbers of primitives in Fig. 21, offering a more intuitive illustration of how the StructMaps' complexity varies. In addition, we compute the distribution of the number of topological holes in the StructMaps and present the results in Tab. 2, illustrating the diversity of topology of the StructMaps in our dataset.

Table 2: The distribution of the number of topological holes in the StructMaps.

| Number of Topological Holes | 0 | 1 | 2 | 3 | 4 | $\geq 5$ |
|---|---|---|---|---|---|---|
| Percentage (%) | 28.4 | 23.1 | 16.8 | 18.9 | 8.5 | 4.3 |

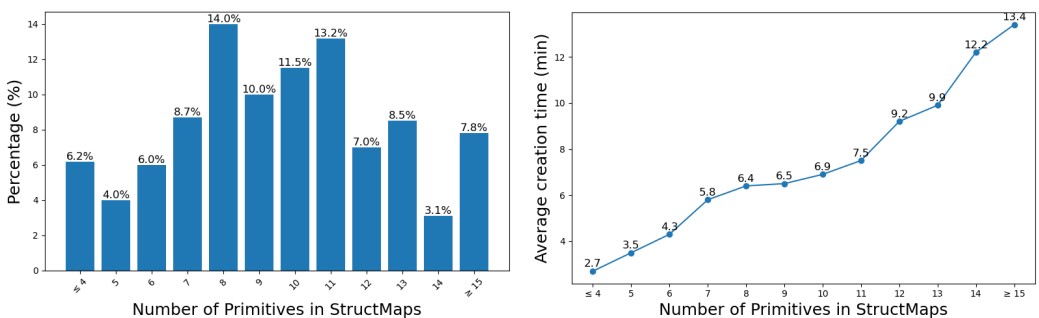

Figure 20: **[Left]** Distribution of the number of primitives in StructMap in our dataset. **[Right]** Average creation time for StructMaps with different numbers of primitives.

| number of primitives | <5 | 5-6 | 7-8 | 9-10 | 11-12 | 13-14 | >15 |
|---|---|---|---|---|---|---|---|
| StructMap | | | | | | | |

Figure 21: Representative examples of StructMaps with different numbers of primitives.

## A.11 THE EFFECTIVENESS OF EACH COMPONENT IN IMAGE GENERATION ALGORITHM

To further evaluate the contribution of each component in our algorithm, we examine four component configurations: (i) image generator only, (ii) combining the image generator with the condition augmentation module, (iii) combining the image generator with the structural consistency discriminator, and (iv) complete algorithm with all three components.

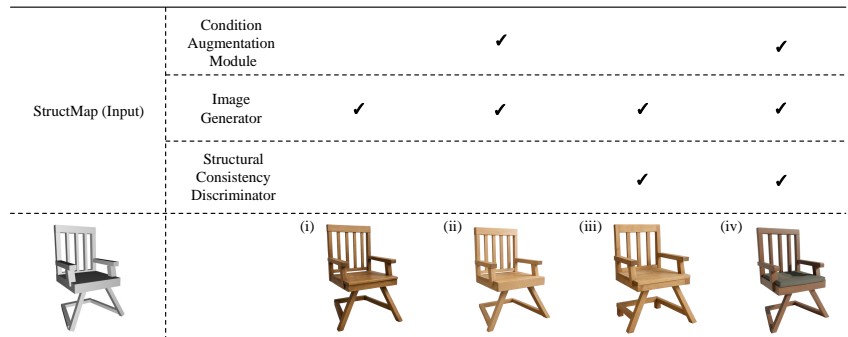

Figure 22: Illustrative example of the generated images with different component configurations: (i) image generator only, (ii) image generator + condition augmentation module, (iii) image generator + structural consistency discriminator, and (iv) complete algorithm.

An illustrative example is presented in Fig. 22. From Fig. 22, We observe that the condition augmentation module corrects the leg misalignment in the generated results, while the structural consistency discriminator rectifies the number of vertical slats on the chair back. These findings demonstrate that each component plays an essential role in achieving high-quality, structure-grounded image generation. We also conducted a quantitative evaluation for each configuration, and the results are shown in Tab. 3. From Tab. 3, we can observe that the Condition Augmentation Module and the Structural Consistency Discriminator lead to substantial improvements in structural consistency (MOS-A), while contributing only modest gains in image quality (FID, MOS-R). This aligns with the design objective of these two components and demonstrates their key role in maintaining structural consistency.

Table 3: Quantitative comparison across different component configurations.

| Component Configurations | FID↓ | MOS-R↑ | MOS-A↑ |
|---|---|---|---|
| image generator only | 39.08 | 4.39 | 4.03 |
| image generator + condition augmentation module | 38.76 | 4.46 | 4.21 |
| image generator + structural consistency discriminator | 38.87 | 4.53 | 4.37 |
| the full algorithm | **38.61** | **4.65** | **4.56** |

## A.12 GENERATION WITH DIFFERENT STYLE LINEARTS

To further compare our method with lineart-based approaches, we additionally created two different styles of linearts as generation conditions: 1) linearts with enhanced visual appeal, inspired by Lineart (Wang et al., 2024a), which provides a highly visually appealing lineart style; 2) linearts that directly trace the geometric components in the StructMaps. The new linearts and their generated results are shown in Fig. 23. From Fig. 23, we can observe that the images generated from the three different styles of lineart achieve roughly comparable quality to those generated with the original lineart in terms of realism and structural consistency. Moreover, their overall quality is generally inferior to that of our method, with only a few GPT-4o-based generations achieving comparable performance. In addition, creating these two new types of lineart takes, on average, more than 30 minutes, imposing a substantial burden on the content creation process.

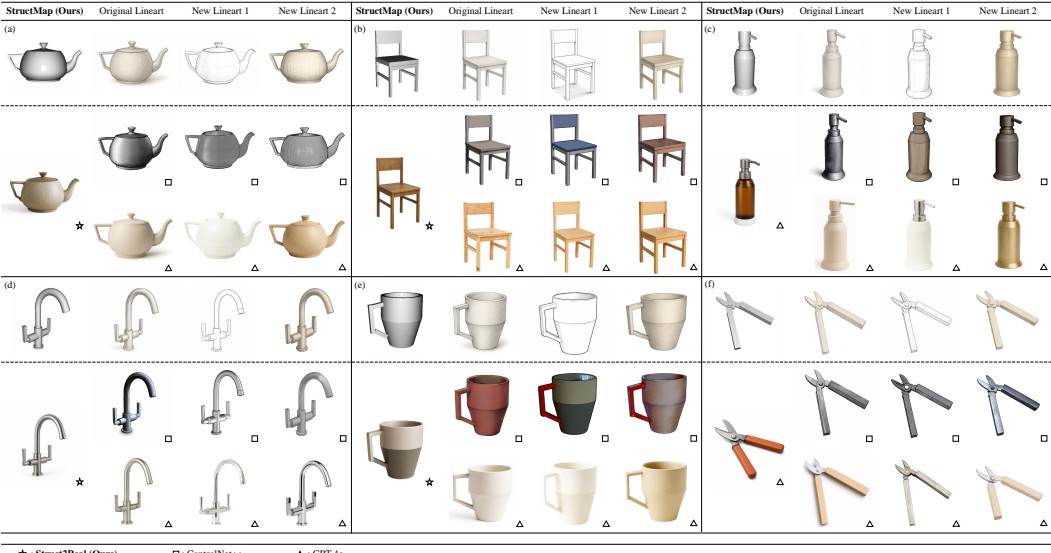

Figure 23: Qualitative results across different style linearts, including: 1) Original Lineart: linearts used in our experiment in the main paper; 2) New Lineart 1: linearts with enhanced visual appeal; 3) New Lineart 2: linearts that directly trace the geometric components in the StructMaps. For each sub-panel, the top row shows the input conditions for each method, and the bottom two rows show the object images generated by different method. The icon in the bottom-right corner of each image indicates the method used to generate it, and the meaning of each icon is explained in the legend at the bottom.

## A.13 COMPARISON OF SKETCHES UNDER DIFFERENT TIME BUDGETS

To further compare StructMaps and sketches in terms of creation efficiency, we created sketches under different time budgets and used the MLLM to generate images from each of them. The results are shown in Fig. 24. Sketches created under a time budget of less than 1 min only roughly outline the object; sketches created with a budget of less than 4 min have clear and undistorted contours, and require correctly handling perspective, which typically involves multiple revisions and fine adjustments; sketches produced under an 8 min budget ensure fully regular edges (e.g., straight lines and ellipses), requiring careful creation with the aid of drawing tools; and with unlimited time,

sketches can incorporate shading and lighting effects to further enhance three-dimensionality. As shown in Fig. 24, only sketches created with a time budget of over 4 minutes are able to achieve structural alignment comparable to StructMaps.

| Creation Budget (min) | 3 (Ours) | <1 | < 4 | < 8 | ∞ |
|---|---|---|---|---|---|
| Input | | | | | |
| Generated Image | | | | | |

Figure 24: Sketches created under different time budgets and corresponding results.

## A.14 NUMBER OF ITERATIONS IN ALGORITHM, API RESPONSE TIME AND FAILURE CASES

To further illustrate the practical behavior of our algorithm, in Fig. 25-Left we present the distribution of the iteration counts of the Structure Consistency Discriminator when generating images using our dataset (We set the maximum number of iterations to 5). As shown in Fig. 25-Left, the vast majority of examples converge in only 1-2 iterations, and the overall convergence success rate reaches 99.7%. Furthermore, we report in Tab. 4 the average API response time for each module of our image generation algorithm.

Table 4: The average API response time for each module of our image generation algorithm.

| Algorithm Component | API Response Time (s) |
|---|---|
| Condition Augmentation Module | 13.23 |
| Image Generator | 56.03 |
| Structural Consistency Discriminator | 9.51 |

In addition, Fig. 25-Right presents several failure cases. For example, in (a), the small sphere at the lower end of the globe's axis is omitted during generation, and in (b), the cabinet's rotation angle is incorrectly produced. These issues may arise because certain components in the image are too small or the overall layout is overly cluttered. We plan to further improve the performance of our method in future work.

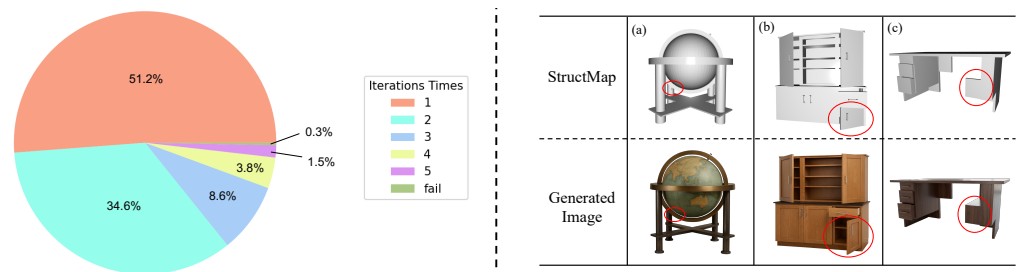

Figure 25: **[Left]** Distribution of the iteration counts of the Structure Consistency Discriminator. **[Right]** Several failure cases.

## A.15 GENERATING MULTI-VIEW IMAGES WITH STRUCTMAP

Our method supports generating consistent multi-view images of an object. Specifically, once a StructMap for an object is created, we can render StructMap images from different viewpoints and

use each of them as a condition to generate multi-view images of the object with our image generation algorithm. Notably, by leveraging simple prompt adjustments along with the contextual memory capability of the MLLM, we can ensure that the generated images maintain a consistent appearance across viewpoints. We present several examples of generating multi-view images in Fig. 26.

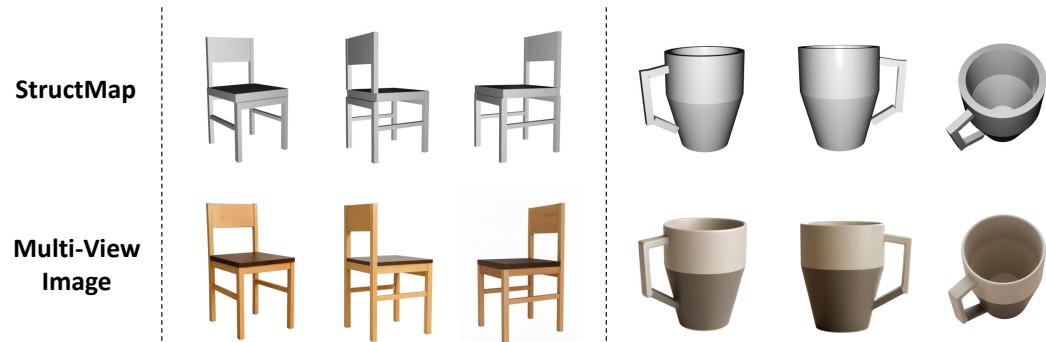

Figure 26: Several examples of generating multi-view images.

### A.16 GENERATING OBJECT IMAGES UNDER DIFFERENT LIGHTING CONDITIONS

Based on our method, we can obtain images of the same object under different lighting conditions, which can be achieved simply by modifying the prompt, and we provide an example in Fig. 27.

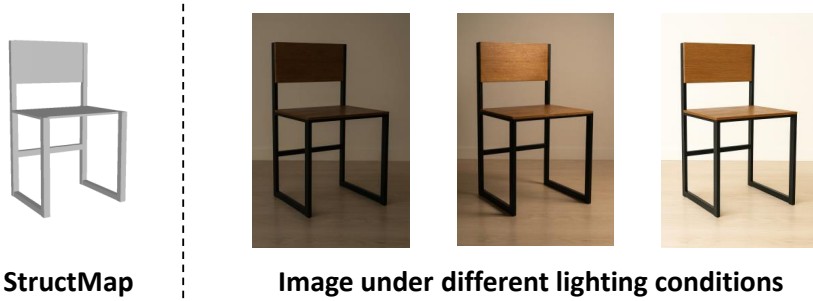

Figure 27: An example of generating images of the same object under different lighting conditions.

### A.17 GENERATING OTHER CATEGORIES OF IMAGES

To further showcase the broader potential of our method, we conducted additional generation experiments, including: 1) generating articulated objects, 2) generating non-rigid objects, and 3) generating multi-object scenes. Specifically, for articulated objects, we create structural conditions of the various articulated states of the same object by modifying the position and rotation parameters of the articulated components within a StructMap; for non-rigid objects, we deform a rigid StructMap according to physically plausible rules in Blender to obtain StructMaps that reflect flexible shapes; and for multi-object scenes, we create the structural condition by placing the StructMaps of different objects at their intended positions in the scene. Once these StructMap conditions are created, we feed each of them into our generation algorithm to synthesize the corresponding photorealistic images. We present several examples of these results in Fig. 28.

### A.18 EXPLORATION OF GENERATING NOVEL CONCEPT OBJECTS

We also applied our method to generate several novel concept objects that do not exist in the real world, and present an example in Fig. 29.

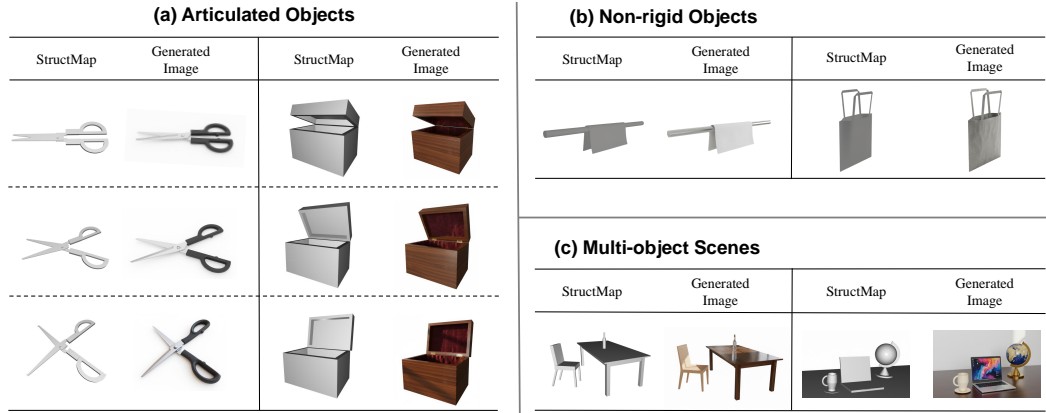

Figure 28: Examples of generating other object types, including: 1) generating articulated objects, 2) generating non-rigid objects, and 3) generating multi-object scenes.

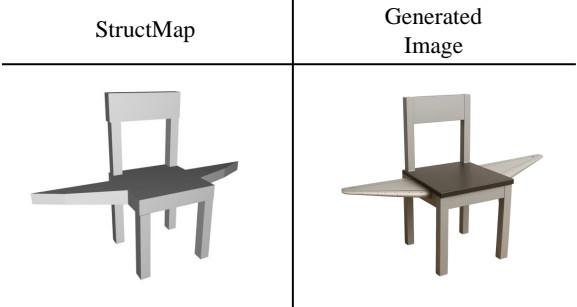

Figure 29: An example of generating novel concept objects. We attempted to generate a "flying chair", which is a chair equipped with wings.

