# OpenReview forum: "Struct2Real: A Systematic Framework for Accurate and Efficient Structure-Grounded Object Image Generation"
_ICLR.cc/2026/Conference — Submitted to ICLR 2026_

### Official Review · Reviewer_Bgc9 · 2025-10-30

**Soundness:** 4
**Presentation:** 3
**Contribution:** 3
**Rating:** 8
**Confidence:** 3

**Summary:**

This paper introduces Struct2Real, a novel framework for generating photorealistic object images under stringent structural constraints, specifically preserving object topology and spatial layout. The framework consists of two main modules: 1) A structure modeling system centered around StructMap, an explicit 3D representation composed of geometric primitives that encode the object's topology and spatial layout. 2) A modular image generation algorithm leveraging Multimodal Large Language Models (MLLMs). Extensive qualitative and quantitative experiments demonstrate that Struct2Real surpasses text, lineart, and scribble-based controllable generation methods in both visual realism and structural fidelity, while requiring lower creation effort.

**Strengths:**

1. Struct2Real introduces a cognitively inspired, part-based 3D abstraction framework named StructMap, which models objects through geometric primitives and their spatial relationships. The design achieves a strong balance between expressiveness and usability, enabling users to specify complex structures intuitively. Moreover, the authors provide a visual design and interaction interface that allows users to construct and inspect StructMaps directly, greatly enhancing interpretability and accessibility.

2. The integration of the StructMap representation with the structure-consistency feedback loop offers a principled mechanism to enforce explicit topology and spatial layout. This feedback-driven process effectively guarantees structural fidelity even under complex geometric configurations, overcoming a key limitation faced by traditional sketch- or layout-based approaches.The modular condition-augmentation and consistency-discriminator design exploits MLLM reasoning to maintain geometric correctness.

3. The overall framework is modular, combining condition augmentation, image generation, and consistency discrimination into a unified process. By leveraging the reasoning capability of multimodal large language models, the system maintains high geometric consistency while synthesizing photorealistic textures, achieving both controllability and realism in structure-grounded image generation.

**Weaknesses:**

1. The method relies heavily on commercial MLLMs such as GPT-4o, which raises reproducibility concerns and may limit open evaluation.

2. The dataset covers only about 30 object categories with 3,000 samples, which restricts scalability and domain generalization. Broader testing on complex scenes or organic, non-rigid objects would strengthen the conclusions.

3. The StructMap representation is inherently constrained by its predefined primitive library. The current examples focus mainly on simple mechanical or geometric shapes, leaving its expressive capacity for more complex structures insufficiently demonstrated.

4. The proposed feedback loop requires multiple rounds of MLLM inference, which could introduce significant computational overhead. A quantitative analysis of runtime and system cost would clarify the method’s practical feasibility.

5. The quantitative evaluation focuses mainly on comparisons with large text-to-image models. It should include comparisons with other structure-aware or 3D-conditioned generation methods that address similar tasks.

**Questions:**

1. Could StructMap be automatically learned or inferred from existing 3D data or multi-view images, rather than being manually constructed?

2. How scalable is the StructMap creation process when modeling complex or deformable objects such as plants, animals, or articulated human figures?

3. What is the computational cost of the full feedback loop? Please quantify the average number of iterations required by the discriminator for convergence.

4. Could the authors quantify the contribution of each component in the modular pipeline through ablation or error propagation analysis?

5. How well does Struct2Real generalize across viewpoints or lighting conditions? Can a single StructMap be used to render consistent multiview outputs, potentially linking to 3D-aware diffusion pipelines?

---

> ### Author Response · Authors · 2025-11-27
>
> We sincerely appreciate the reviewer’s recognition of our work, as well as the many valuable suggestions provided. To better address the reviewer’s questions and further improve our paper according to the reviewer’s suggestions, we conducted many supplementary experiments and carefully revised the paper, which resulted in a slight delay in our response. We kindly ask for the reviewer’s understanding, and  our responses to the reviewer’s comments are detailed below:
>
> **W1: Reproducibility concerns and limitation in open evaluation.**
> Thank you very much for this comment. Our method can be easily adopted or reproduced by others, although it does require some financial costs. Yet, given the rapid development of MLLMs, we believe that open-source MLLMs with strong capability will emerge in the near future, which will make it easier for the community to experiment with our work.
>
> **W2: Scalability and domain generalization of the dataset.**
> Thank you for the helpful suggestion. In Appendix A.17, we have added several examples illustrating the generation of multi-object scenes or non-rigid objects, which demonstrate that our method can extend to these types of images.
>
> **W3: Expressive capacity for more complex structures.**
> Thank you very much for this comment. StructMaps are capable of representing more complex object structures, because our geometric primitives come with **rich parameterization** and are designed to **align closely with the structural patterns of real-world objects**. We have added additional examples with more complex structures in appendix A.8, which demonstrate that our method can also handle complex structures effectively.
>
> **W4: Runtime and system cost.**
> Thank you for the valuable suggestion. In appendix A.14, we have included detailed statistics on the number of iterations required across our experimental dataset, and the average runtime of our experiments is 2 min 04 s.
>
> **W5: Comparisons with other structure-aware or 3D-conditioned generation methods.**
> Thank you for the suggestion. However, to the best of our knowledge, there is currently no structure-aware or 3D-conditioned method capable of accomplishing our task, and therefore we did not include such comparisons.

---

> ### Author Response · Authors · 2025-11-27
>
> **Q1: Could StructMap be automatically learned or inferred from existing 3D data or multi-view images?**
> Thank you very much for your question. In this paper, StructMap cannot be fully automatically extracted from existing 3D data or multi-view images, yet we believe that it is feasible. Here, we provide one possible direction: First, we can perform semantic segmentation to decompose a complex object into multiple components. Next, we leverage VLMs to infer the required types and quantities of geometric primitives for each part, and then instantiate each part using the selected primitives by regressing their parameters. Finally, we assemble all components to produce the corresponding StructMap. We appreciate your meaningful question, and we plan to explore it in our future work.
>
> **Q2: How scalable is the StructMap creation process when modeling complex or deformable objects such as plants, animals, or articulated human figures?**
> Thank you very much for your meaningful question. We suggest that StructMap can be extended to complex or deformable objects such as plants, animals, or articulated human figures by designing a set of geometric primitives tailored to their common structural characteristics, such as using branch- and spline-based primitives to model the structure of trees, bushes, ferns, or cacti[1]. In this paper, we first validate the feasibility of our approach on the more commonly used daily objects, and we plan to extend our method to plants, animals, and articulated human figures in future work.
>
> **Q3-1: What is the computational cost of the full feedback loop?**
> The full feedback loop requires about 1–2 minutes of API response time, and the overall image generation pipeline takes about 2–3 minutes in total.
>
> **Q3-2: Average number of iterations.**
> The average number of iterations is 1.69, and in appendix A.14 we have added detailed statistics on the number of iterations required across our experimental dataset.
>
> **Q4: Quantify the contribution of each component in the modular pipeline.**
> Thank you for the valuable suggestion. We have added a quantitative ablation study in Appendix A.11 to evaluate the contribution of each component in the generation algorithm, and the results further confirm the effectiveness of each module.
>
> **Q5: How well does Struct2Real generalize across viewpoints or lighting conditions? Can a single StructMap be used to render consistent multiview outputs?**
> Thank you for your meaningful questions. Our method is able to generate consistent images across different viewpoints or lighting conditions. Specifically, once a StructMap for an object is created, we can render StructMap images from different viewpoints and use each of them as a condition to generate multi-view images of the object with our image generation algorithm. In Appendix A.15, we have added examples of generating realistic images from StructMap images rendered from different viewpoints. And in Appendix A.16, we also provide examples of the same object generated under different lighting conditions, which can be achieved simply by modifying the prompt.
>
> [1] A. Joshi, B. Han, J. Nugent, M. G. Saez-Diez, Y. Zuo, J. Liu, H. Wen, S. Alexandropoulos, K. Kayan, A. Calveri, T. Sun, G. Liu, Y. Shao, A. Raistrick, and J. Deng, “Procedural Generation of Articulated Simulation-Ready Assets,” arXiv preprint arXiv:2505.10755, 2025.

---

### Official Review · Reviewer_eGEB · 2025-10-31

**Soundness:** 4
**Presentation:** 3
**Contribution:** 3
**Rating:** 6
**Confidence:** 4

**Summary:**

This paper proposes Struct2Real, a framework for structure-grounded object image generation. It introduces a new 3D structural representation called StructMap, composed of geometric primitives encoding object topology and spatial layout. The framework includes:
(1) a structure modeling system allowing users to assemble StructMaps via an interactive interface, and
(2) an image generation algorithm that combines StructMaps with multimodal large language models (MLLMs) to produce photorealistic images faithful to the provided structure.
Experiments compare Struct2Real with text, lineart, and scribble-based conditioning under various baselines (OmniGen, ControlNet++, T2I-Adapter, etc.), evaluated by FID and human MOS ratings.

**Strengths:**

1. This work presents a clear motivation for structure-grounded control in object image generation, supported by cognitive inspiration (Recognition-by-Components theory).


2. This work introduces StructMap, a clean, interpretable representation that enables explicit structural input.

3. Demonstrates broad evaluation—multiple baselines, diverse conditioning modalities, and human studies on both realism and structural alignment.

4. Visual examples are compelling and show genuine improvements in both realism and structure control.

**Weaknesses:**

1. The paper relies solely on MOS for assessing structure–image alignment (Sec. 4.1 → A.2.5). This weakens objectivity.


2. In Fig. 4, the iterative consistency-checking process is central, yet there is no evidence on iteration counts, failure cases, or convergence stability.

3. The claimed “3000 samples, 30 categories” dataset is newly built but lacks public availability or validation diversity. Examples of StructMap complexity (number of primitives, topology variety) are missing.

4. The proposed StructMap indeed provides a more accurate and explicit representation of object structures. However, since it relies on specific geometric priors and requires a dedicated software interface for creation, its applicability remains somewhat limited in terms of flexibility and accessibility, especially when compared to more lightweight and widely usable inputs such as text or scribble conditions.

**Questions:**

1. How is the “structure consistency discriminator” implemented—purely via LLM reasoning or also via visual feature comparison? How many regeneration rounds are typically required before convergence?

2. Could StructMap be automatically extracted from existing CAD or mesh data? If yes, how scalable is the user-creation interface beyond toy-level examples?

3. Have you evaluated Struct2Real on multi-object scenes or non-rigid categories? Does the geometric-primitive abstraction generalize beyond single rigid objects?

4. Is there a possibility to learn a StructMap-to-latent alignment (e.g., via adapter or encoder) instead of relying entirely on prompting?

5. Will you release the structure-prior dataset and interface tools for reproducibility?

---

> ### Author Response · Authors · 2025-11-27
>
> We are grateful for the reviewer’s positive assessment of our work, and the reviewer’s suggestions have been extremely helpful for further improving the paper. To better address the reviewer’s questions and further improve our paper according to the reviewer’s suggestions, we conducted many supplementary experiments and carefully revised the paper, which resulted in a slight delay in our response. We kindly ask for the reviewer’s understanding, and our detailed replies to the reviewer’s comments are provided below:
>
> **W1: Paper relies solely on MOS for assessing structure-image alignment.**
> Thank you very much for this comment. In Appendix A.2.4, we have provided a detailed explanation of why we did not adopt common evaluation metrics such as SSIM or LPIPS in our original submission. Specifically, these metrics are not designed to evaluate the alignment of topology and spatial layout, but rather focus on pixel-level or feature-level alignment. Our work places a significant emphasis on the alignment of topology and spatial layout, which go beyond pixel or feature accuracy and involve a more abstract understanding of topological configurations and spatial organization.
> For our structure-controlled image generation task, there is currently no suitable objective metric that can effectively measure the consistency between the generated image and the given structural configuration. Therefore, we opted for human evaluation instead. We believe your question is highly meaningful, and we will continue exploring suitable metrics in future work to further demonstrate the effectiveness of our method.
>
> **W2: Iteration counts, failure cases, or convergence stability of the consistency-checking process.**
> In Appendix A.14, we have added detailed statistics on the number of iterations required across our experimental dataset. In practice, we cap the number of iterations at 5, and in our dataset, 99.7% of the cases successfully converge within 5 iterations, and we provide several examples of failure cases in Appendix A.14.
>
> **W3: Dataset lacks public availability or validation diversity.**
> Thank you very much for this comment. We will release our dataset as soon as the paper is published. In addition, we have added detailed statistics in Appendix A.10 on both the number of primitives in the StructMaps and their topological hole counts. These statistics clearly demonstrate the diversity of our dataset in terms of structural complexity.
>
> **W4: StructMap is limited in flexibility and accessibility.**
> Thank you very much for this comment. Our task is structure-controlled image generation, and based on this task, we respond from the following perspectives:
> 1) This task places high demands on the structural expressiveness and accuracy of the condition.   StructMap meets these requirements, but more flexible and accessible conditions such as text or scribbles cannot satisfy the needs of our task, as shown in Fig.5 of the main paper.
> 2) Under the above requirements, our method is already highly flexible and accessible, especially campared with lineart. Specifically, the creation of a StructMap operates in 3D space, which places a relatively low demand on the user's spatial imagination, and if the user wishes to modify the structure, they only need to add, remove, or adjust the corresponding primitives. In contrast, translating a complex 3D structure into a 2D lineart while preserving correct spatial relationships is challenging for non-expert users, as drawing accurate proportions and perspective of a 3D object is difficult. And once the lineart is completed, modifying the structure becomes very difficult.

---

> ### Author Response · Authors · 2025-11-27
>
> **Q1-1: How is the "structure consistency discriminator" implemented?**
> Our structure consistency discriminator is implemented through MLLM reasoning.
>
> **Q1-2: How many regeneration rounds are typically required before convergence?**
> Before convergence, it requires an average of 1.69 iterations, and we have added detailed statistics on the number of iterations required across our experimental dataset in Appendix A.14.
>
> **Q2: Could StructMap be automatically extracted from existing CAD or mesh data?**
> Thank you very much for your question. In this paper, StructMap cannot be fully automatically extracted from existing CAD or mesh data, yet we believe that it is feasible. Here, we provide one possible solution: First, we can perform instance segmentation to decompose a complex object into multiple parts. Next, we leverage VLMs to infer the required types and quantities of geometric primitives for each part, and then instantiate each part using the selected primitives by regressing their parameters. Finally, we assemble all components to produce the corresponding StructMap. We appreciate your meaningful question, and we plan to explore it in our future work.
>
> **Q3: Evaluate Struct2Real on multi-object scenes or non-rigid categories?**
> Thank you very much for your meaningful question. Our method is able to generalize to multi-object scenes and non-rigid categories, and in Appendix A.17, we have added several examples illustrating the generation of multi-object scenes or non-rigid categories, which demonstrate that our method can extend to these types of images.
>
> **Q4: Is there a possibility to learn a StructMap-to-latent alignment?**
> Thank you very much for your meaningful question, and we consider that it is possible to learn a StructMap-to-latent alignment. Yet, we have not explored this direction in depth in our paper, as MLLMs have demonstrated strong capabilities in understanding structural and semantic information in images and generating high-quality visual content, and our current focus is primarily on leveraging and activating the capabilities of MLLMs for our structure-grounded image generation task. We will further investigate this direction in future work.
>
> **Q5: Will you release the structure-prior dataset and interface tools for reproducibility?**
> Yes, we will release both the structure-prior dataset and the interface tools as soon as the paper is published to support reproducibility.

---

### Official Review · Reviewer_zjTn · 2025-11-02

**Soundness:** 3
**Presentation:** 1
**Contribution:** 2
**Rating:** 4
**Confidence:** 3

**Summary:**

The paper introduces Struct2Real, a framework for structure-grounded object image generation that leverages explicit 3D structural priors (called StructMap) and integrates multimodal large language models (MLLMs) for photorealistic image synthesis under topology and spatial layout constraints. The work addresses a long-standing challenge in controllable generation — maintaining structural fidelity while achieving high realism. The proposed system includes (1) a structure modeling interface for creating StructMaps, (2) a condition augmentation and reasoning pipeline using MLLMs, and (3) a structure-consistency discriminator for iterative refinement. Experiments show consistent improvements over text, lineart, and scribble-based baselines in both realism (FID, MOS-R) and structure alignment (MOS-A).

**Strengths:**

1. a 3D composition of geometric primitives encoding topology and layout is conceptually elegant and practically useful. It provides a middle ground between coarse 2D conditions (e.g., lineart) and complex 3D CAD models.
2. qualitative evaluations shows good performance in both realism and structure preservation.

**Weaknesses:**

1. while StructMap is new, the image generation algorithm primarily relies on prompting existing MLLMs (e.g., GPT-4o). There is little discussion of model-specific innovations or learnable components beyond prompt design.

**Questions:**

1. Line 192-193 "these conditions are often coarse-grainedor ambiguous,making it difficult to accurately reflect the object’s structure" do you have an exmple? can you explain?
2. the paper talk a lot on 3D structure, it can generate novel view?
3. can you suggest new way of controllability? in manner no one done before?
4. Could Struct2Real generalize to articulated or deformable objects? new concepts objects?

---

> ### Author Response · Authors · 2025-11-27
>
> Thank you very much for your time and for the valuable feedback provided, which has helped us improve the paper. To better address the reviewer’s questions and further improve our paper according to the reviewer’s suggestions, we conducted many supplementary experiments and carefully revised the paper, which resulted in a slight delay in our response. We kindly ask for the reviewer’s understanding, and our responses are as follows:
>
> **W1: Little discussion of model-specific innovations or learnable components.**
> Thank you very much for this comment. We humbly suggest that the little discussion of model-specific innovations or learnable components does not diminish the novelty of our image generation algorithm, for the following reasons:
> 1) As pointed out in the comment, we introduce a new type of condition, StructMap, which accurately represent object topology and spatial layout. To leverage the comprehensive structural information in the StructMap, we need to design a corresponding algorithm that generates photorealistic object images while preserving the structural configuration it encodes. **The development of such an algorithm is itself a novel contribution.**
> 2) Traditional approaches which require model-specific training, such as those based on ControlNet or LoRA, mainly perform pixel-level alignment, and thus struggle to generate natural geometric details and smooth connections between primitives, resulting in poor realism in the generated images (as shown in Fig.5 of the main paper). In contrast, MLLMs possess strong capabilities in understanding structural and semantic information in images and generating high-quality visual content, making them a promising choice for achieving our image generation objectives. Therefore, we chose to activate and leverage the powerful capacities of MLLMs. **This exploration of the potential of MLLMs for controllable image generation** also constitutes a novel contribution of our work.

---

> ### Author Response · Authors · 2025-11-27
>
> **Q1: Explain "these conditions (including text, semantic layouts, and pose keypoints) are often coarse-grained or ambiguous".**
> As stated in lines 198-201 of the main paper (new version), text, semantic layouts, and pose keypoints provide only coarse-grained descriptions of an object's structure, and we have added an example in Appendix A.1 to illustrate this more concretely. Specifically, textual descriptions are clearly insufficient for precisely conveying the structure of a 3D object, especially regarding the size, spatial layout and the connections between different components. Semantic layouts can only specify the semantic category of each spatial region, but they are still unable to precisely capture the topology and shape of an object, and their characterization of the spatial layout is also insufficiently detailed. Finally, pose keypoints only provide information about topological connectivity and cannot capture the shape or size of each component.
>
> **Q2: Can it generate novel view?**
> Thank you very much for your practical question. Yes, our method is capable of generating images of an object from different viewpoints. Specifically, once a StructMap for an object is created, we can render StructMap images from different viewpoints and use each of them as a condition to generate multi-view images of the object with our image generation algorithm. In Appendix A.15, we provide several examples of generated multi-view images, which show that our method is able to produce consistent multi-view object images.
>
> **Q3: New way of controllability**
> Thank you very much for your question. Yet, we found it to be slightly ambiguous, so we respectfully clarify that our understanding of the question is “Have you proposed a new form of controllability, in a way that has not been explored before?”, and we provide our response accordingly as follows:
> Our method provides a new form of controllability, as it allows users to control the **topology and spatial layout** of the generated object **within a 3D space** while preserving the **realism** of generated object images. Prior methods neither support structural design in 3D space (e.g., using text or sketches as conditions), resulting in an inaccurate or incomplete representation of 3D structure, nor can they maintain the realism of the generated images when controlling an object's topology and spatial layout (e.g., limited to pixel-level control, which leads to low realism, as discussed in W1).
> If our understanding is inaccurate, we would greatly appreciate your clarification and address your concern as soon as possible.
>
> **Q4-1: Could Struct2Real generalize to articulated or deformable objects?**
> Thank you very much for your meaningful question. Yes, Struct2Real is able to generalize to articulated or deformable objects, and in Appendix A.17, we have added several examples illustrating the generation of articulated or deformable objects, which demonstrate that our method can extend to these types of objects.
>
> **Q4-2: New concepts objects?**
> Thank you very much for your question. Yes, we can generate images of new concepts objects, and we have added an example in Appendix A.18. If our understanding of "new concepts objects" is inaccurate, we would greatly appreciate your clarification and address your concern as soon as possible.

---

### Official Review · Reviewer_YmnD · 2025-11-03

**Soundness:** 3
**Presentation:** 3
**Contribution:** 3
**Rating:** 4
**Confidence:** 4

**Summary:**

This paper addresses the problem of generating high-quality, photorealistic object images under fine-grained structural constraints—specifically preserving an object’s topology and spatial layout. Unlike existing controllable image generation methods that either offer only coarse structural guidance or require high professional skills, this work focuses on balancing precise structural control, visual realism, and user-friendliness, ensuring low effort for users while maintaining structural fidelity.
To tackle this challenge, the authors propose Struct2Real, a two-module framework integrated with multimodal large language models (MLLMs), featuring:
- a structure modeling system centered on "StructMap"—a 3D structural representation built from geometric primitives and their adjustable properties . StructMap explicitly encodes an object’s topology and spatial layout, and the accompanying interactive interface lets users assemble StructMaps without specialized skills;
- a modular image generation algorithm that works with MLLMs to translate StructMaps into photorealistic images, including three core components:
  1. a Condition Augmentation Module that converts StructMap images into detailed textual descriptions to bridge the gap between structural inputs and language-preferred generative models, highlighting key structural details like part count, connections, and spatial arrangement;
  2. an Image Generator that uses MLLMs to produce images conditioned on both StructMap images and their textual descriptions, preserving structural constraints while adding realistic textures, materials, and fine details. Users can also add optional style prompts to customize appearance;
  3. a Structure Consistency Discriminator that forms a feedback loop—MLLMs compare generated images with StructMaps to identify structural inconsistencies, provide reasoning for mismatches, and guide the generator to regenerate until the image aligns with the input structure;
- compatibility with multiple MLLMs to ensure generality across different multimodal model backends.
Experiments on a manually constructed "structure-prior dataset" show this method outperforms state-of-the-art baselines: it delivers better image realism and structural alignment than text, lineart, and scribble-based methods; StructMap balances accessibility and performance; and ablation studies confirm the framework’s generality across MLLMs and the necessity of each component for strong results.

**Strengths:**

- The paper is well written and easy to follow. The figures are clear, visually appealing, and effectively support the explanations.
- The experimental results are impressive and convincingly demonstrate the effectiveness of the proposed framework.
- The overall approach is conceptually sound and logically consistent—it makes good sense and aligns well with the problem formulation.

**Weaknesses:**

- Although the authors claim that their 3D structural representation is easy to obtain, this holds mainly for objects with relatively simple geometry. The method becomes less practical for complex shapes, which limits its applicability to more intricate or detailed structures.
- While the system design is reasonable and well engineered, it lacks strong conceptual novelty. The work feels more like a comprehensive engineering integration rather than a fundamentally new algorithmic contribution.
- The proposed pipeline, though effective, seems somewhat over-engineered for the task. The overall process could be viewed as unnecessarily complicated relative to the problem’s scale.
- Some of the line-art examples used in the experiments appear to be of relatively low quality. When high-quality line-art conditions are used, the performance gap between baseline methods and the proposed approach becomes much smaller, which raises questions about the fairness of the dataset and evaluation setup.

**Questions:**

1. The generated objects in your paper are mostly structurally simple. Could you provide more challenging examples that better demonstrate the capability and generalization of your method? While your structural control is indeed impressive, the simplicity of the examples limits the practical significance. Designing such simple shapes (e.g., cups) may not justify the relatively complex pipeline you propose—in some cases, manually sketching might even be more efficient.
2. In Figure 5, don’t you think some of your geometry conditions are excessively detailed, while certain line-art conditions appear overly coarse? Regardless of how difficult these inputs are to obtain, could you show what would happen if one directly traced the geometric components to create line-art conditions? This would clarify whether the gap stems from representation quality rather than the generation method itself.
3. Could you provide a quantitative estimate of the time required to construct geometric conditions? For instance, how long would it take to build the geometry condition for a simple object like a cup? And how would this time scale with more complex shapes—does the modeling effort grow significantly with geometric complexity?

---

> ### Author Response · Authors · 2025-11-27
>
> We sincerely appreciate the reviewer for the time devoted to reviewing our work and the insightful suggestions provided. The feedback has been highly beneficial in helping us refine our paper. To better address the reviewer’s questions and further improve our paper according to the reviewer’s suggestions, we conducted many supplementary experiments and carefully revised the paper, which resulted in a slight delay in our response. We kindly ask for the reviewer’s understanding, and our detailed responses are provided below:
>
> **W1: Less practical for complex shapes.**
> Thank you very much for this comment. We respectfully argue that our StructMap is also practical for objects with complex structures. For complex objects, easily created conditions such as text or scribbles are impractical, as they cannot accurately express the complex structures. And creating a commonly used condition such as lineart for a complex object is inherently challenging[1, 2], often requiring more than 30 minutes to produce. In contrast, creating a StructMap for a complex object typically takes less than 15 minutes, indicating that our method is already **highly practical for this challenging task**. We have added examples of creating StructMaps for complex objects in Appendix A.9 to demonstrate the practicality of our method.
> We appreciate your meaningful comment, and during actual usage, we have found that the efficiency of creating StructMaps can be further improved by extending the functionality of the StructMap Design Interface. For example, we can design commonly used combinations of these primitives, enabling users to instantiate multiple primitives simultaneously, or enable users to reuse previously created StructMaps as components when building new ones. We have added the relevant content in lines 248-252 of the main paper, and we will include these functionalities in the interface when we release the code.
>
> **W2: Lacks strong conceptual novelty.**
> Thank you for the valuable feedback. We humbly suggest that our method are conceptually novel for the following reasons:
> 1) For modeling object structures, StructMap encodes an object’s topology and spatial layout using a composition of geometric primitives, providing a **conceptual novel way** to efficiently and accurately **represent object structure in 3D space**, and serving as **an effective condition for structure-controlled image generation**. *Existing generation conditions including text, sketches, semantic layouts, and pose keypoints are not able to achieve this*, as discussed in Sec. 4.2 of the main paper and Appendix A.1.
> 2) In terms of designing the generation algorithm, we need to leverage and preserve the structural information in the StructMap and transform it into photorealistic images. Traditional methods based on ControlNet or LoRA perform pixel-level alignment, which hinders them from generating natural geometric details and smooth connections between primitives, resulting in poor realism in the generated images (as shown in Fig.5 of the main paper). In contrast, we found that MLLMs possess strong capabilities in understanding structural and semantic information in images and generating high-quality visual content, and thus designed an algorithm that **activates and leverages these capabilities** to generate photorealistic images while maintaining structural consistency. This **exploration of the potential of MLLMs for structure-controlled image generation** is also conceptual novel.
>
> **W3: Pipeline seems over-engineered.**
> Thank you for your concern. We respectfully argue that our design is necessary and does not introduce significant complexity for the following reasons:
> 1) The three components in our image generation algorithm, Condition Augmentation Module, Image Generator, and Structure Consistency Discriminator, each have their own specific function.
>     * Condition Augmentation Module makes the MLLM accurately **capture structural information**.
>     * Image Generator serves as the **core image synthesis module**, which is indispensable.
>     * Structure Consistency Discriminator **enforces structural consistency** between the MLLM's output image and the input StructMap condition.
>
>     In Appendix A.11, we have added qualitative and quantitative experiments to evaluate the contribution of each component in the generation algorithm, and the results show that each module contributes to improving the quality of the generated images.
>
> 2) In practical usage, our image generation algorithm **does not need to be executed strictly following the full pipeline**. We provide the most rigorous workflow to ensure robustness across the majority of cases. In certain scenarios, users may decide whether to apply the Condition Augmentation Module and the Structure Consistency Discriminator for efficiency considerations.

---

> ### Author Response · Authors · 2025-11-27
>
> **W4: Line-art appear to be of relatively low quality.**
> Thank you very much for this comment. We respectfully clarify that our original lineart was carefully created by professional artists, and we suggest that its quality is already high. Nevertheless, we referred to linearts used in other works and **selected a highly visually appealing style** from Lineart[3]. We then recreated new version of the linearts following this style (which required a longer creation time of over 30 minutes) and added the corresponding results to Appendix A.12. From the results, we can see that the images generated using the new linearts achieve roughly comparable quality to those generated with the original lineart in terms of realism and structural consistency, and neither surpasses the performance of our method. Therefore, we believe that the lineart used in our original experiments is already sufficient to enable lineart–based methods to perform at their best, ensuring **the fairness of our evaluation**.
>
> **Q1-1: Provide more challenging examples.**
> In Appendix A.9, we have added several examples of image generation for more complex objects. These results show that our method can also handle complex structures effectively, demonstrating strong capability and generalization.
>
> **Q1-2: In some cases (e.g., creating cups), manually sketching might even be more efficient.**
> We humbly suggest that for relatively simple structures such as cups, the efficiency of creating a StructMap is comparable to that of creating a useful sketch. To verify this, we created sketches under different time budgets and used them for image generation, and the results are provided in Appendix A.13. As shown, only sketches created with a time budget of 8 minutes or more are able to achieve structural alignment comparable to StructMaps, while creating a StructMap itself takes only about 3 minutes.
>
> **Q2: Line-art conditions appear overly coarse.**
> Thank you very much for your question. We respectfully argue that our lineart is not overly coarse and that our original lineart was carefully created by professional artists. Nevertheless, we followed the suggestion and created a set of linearts that **directly trace the geometric components in the StructMaps** (which required a longer creation time of over 30 minutes), and included the corresponding results in Appendix A.12. As shown in the results, the images generated using the new linearts achieve roughly comparable quality to those generated with the original lineart in terms of realism and structural consistency, and neither surpasses the performance of our method. This indicates that the gap in generation quality **does not stem from any insufficiency in the lineart we provided**.
>
> **Q3-1: Quantitative estimate of the time required to construct geometric conditions.**
> We have presented the average time required to create a StructMap in Tab.1 of the main paper. In addition, we have added more detailed creation times for StructMaps of different complexity levels in Appendix A.10.
>
> **Q3-2: How long would it take to build the geometry condition for a simple object like a cup?**
> Creating a simple StructMap such as a cup requires about 2-3 minutes.
>
> **Q3-3: How would this time scale with more complex shapes?**
> The creation time shows an approximately positive linear correlation with the number of primitives in the StructMap. And based on our statistics in Appendix A.10, the creation time for most complex objects can be kept within 15 minutes.
>
> [1]Edgar Simo-Serra, Satoshi Iizuka, Kazuma Sasaki, and Hiroshi Ishikawa. 2016. Learning to simplify: fully convolutional networks for rough sketch cleanup. ACM Trans. Graph. 35, 4, Article 121 (jul 2016), 11 pages. doi:10.1145/2897824.2925972
> [2]Simo-Serra, E., Iizuka, S., and Ishikawa, H.Real-Time Data-Driven Interactive Rough Sketch Inking. ACM Transactions on Graphics (SIGGRAPH 2018), 37(4), Article 98, 2018. https://doi.org/10.1145/3197517.3201370
> [3] Xi Wang, Hongzhen Li, Heng Fang, Yichen Peng, Haoran Xie, Xi Yang, and Chuntao Li. Lineart: A knowledge-guided training-free high-quality appearance transfer for design drawing with diffusion model. In Proceedings of the Computer Vision and Pattern Recognition Conference (CVPR), pp. 2912–2923, June 2024a.

---

### Author Response · Authors · 2025-12-03
**Review Summary by the Authors (1/2)**

Dear Area Chair,

Thank you very much for your time invested in reviewing our work, and we would like to provide a brief summary of the discussion period in the hope of assisting the AC in their final assessment.

---

## Summary of Our Paper

We propose Struct2Real, a novel framework for **structure-grounded object image generation** that combines explicit structural control with photorealistic generation, consisting of twofold: 1) A novel **structure modeling system** that enables users to create a 3D structural representation named **StructMap**, which can efficiently and accurately represent the **topology and spatial layout** of an object; 2) A modular **image generation algorithm** that activates and leverages the MLLM's superior performance in understanding structural and semantic information in images and generating high-quality visual content, and generate **photorealistic object images** under the **structural constraints** encoded in StructMap.

---

## Summary of the Discussion Period

**Scoring:** We list the reviewers’ scores below. As can be seen, the overall assessment of our paper is positive, with all average scores falling above the median. Specifically, we sincerely appreciate reviewers Bgc9 and eGEB for their strong recognition of our work, as reflected in the high scores they provided across all aspects. And reviewers zjTn and YmnD initially provided a rating of 4. We are grateful for their concerns and suggestions, we have provided detailed responses to address each of their questions and concerns.

| Reviewer | Rating | Confidence | Soundness | Presentation | Contribution |
|:----:|:----:|:----:|:----:|:----:|:----:|
| Bgc9 | 8 | 3 | 4 | 3 | 3 |
| eGEB | 6 | 4 | 4 | 3 | 3 |
| zjTn | 4 | 3 | 3 | 1 | 2 |
| YmnD | 4 | 4 | 3 | 3 | 3 |
| **Average** | **5.5** | **3.5** | **3.5** | **2.5** | **2.75** |


**Strength:** We greatly appreciate the reviewers’ positive assessment of the strengths of our work. Specifically:
- Struct2Real introduces a **cognitively inspired, part-based 3D structural representation** named StructMap. The design is both elegant and effective, achieving a strong **balance between expressiveness and usability**, and enabling users to specify complex structures intuitively. (Reviewer zjTn, eGEB, Bgc9)
- The integration of StructMaps with a novel image generation algorithm provides a principled mechanism for enforcing **topology and spatial layout**, enabling **robust structural fidelity** even under complex geometries, overcoming a key limitation faced by traditional *sketch- or layout-based approaches*. (Reviewer Bgc9)
- Our experimental results are highly compelling, demonstrating **strong visual realism** while maintaining **excellent structural consistency**, and convincingly showing the effectiveness of the proposed framework. (Reviewer YmnD, zjTn, eGEB)
- Our evaluation is comprehensive and broad, covering **multiple baselines, diverse conditioning modalities, and human studies** assessing both realism and structural alignment. (Reviewer eGEB)
- Our paper presents a **clear motivation**, addresses a **meaningful problem**, and proposes a **logical and well-justified solution**. (Reviewer YmnD, eGEB, Bgc9)
- The paper is **well written** and **easy to follow**. (Reviewer YmnD)

---

> ### Author Response · Authors · 2025-12-03
> **Review Summary by the Authors (2/2)**
>
> **Reviewers’ Concerns and Our Responses**: We have carefully responded to all of the reviewers’ concerns and questions, and have added necessary discussions, experiment results, and other revisions to our submission to better address these concerns in the mean time. To concisely highlight the key points, we summarize below the major concerns raised by the reviewers whose ratings were 4 (YmnD, zjTn), along with our responses.
>
> - **Performance on more complex objects (Reviewer YmnD)**: Our dataset already includes StructMaps with a wide range of structural complexities(as shown in Appendix A.10). And to intuitively demonstrate our method’s performance on challenging objects, we have added examples of creating StructMaps with highly complex structures and generating corresponding object images in Appendix A.9. The results demonstrate that StructMap effectively **balances expressive power and efficiency** when representing such complex objects, and that our method **generates photorealistic images with excellent structural fidelity**.
> - **Novelty of our method (Reviewer YmnD, zjTn)**: We have further highlighted the novelty of our method in the revised paper and provided additional clarification to the reviewers regarding its core innovations: 1) StructMap provide a **conceptually novel way** to efficiently and accurately represent object structure in 3D space, serving as an effective **condition for structure-controlled image generation**. 2) Our image generation algorithm activates and leverages the capabilities of MLLM in understanding structural and semantic information in images and generating high-quality visual content, exploring **the potential of MLLMs in controllable image generation**. *We also appreciate reviewer Bgc9’s recognition of the novel design of both our StructMap and generation algorithm, as highlighted in the Strengths section.*
> - **Contribution of each component in image generation algorithm (Reviewer YmnD)**: We previously conducted a component-wise analysis in Appendix A.5.3, and we have added more detailed qualitative and quantitative experiments in Appendix A.11 to further evaluate the contribution of each component in the generation algorithm, and the results show that **each module contributes to improving the quality of the generated images**.
>
> ---
>
> ## Final Summary
>
> We believe our submission makes a meaningful contribution to **structure-conditioned image generation**. Struct2Real introduces a novel structural representation, **StructMap**, that provides an efficient and expressive way to encode object **topology and spatial layout**, and a modular **image generation algorithm** that effectively leverages MLLMs for **photorealistic, structure-aligned** image generation.
>
> We sincerely appreciate the reviewers’ thoughtful evaluation and their recognition of the strengths of our work, including its **well-designed structural representation and generation algorithm, strong experimental results, comprehensive evaluation, and clear motivation**.
>
> Meanwhile, we have provided detailed clarifications to address the reviewers’ concerns such as **the novelty of our method and the component-wise contribution**. And we have incorporated the reviewers’ constructive suggestions, such as **evaluating performance on highly complex objects and examining generalization ability**, by adding corresponding supplementary experiments and revising the paper to further strengthen our submission.
>
> We hope that this summary, together with our detailed responses, supplementary experiments, and paper revisions, can be of assistance in forming a well-informed final assessment of our work.
>
>
> Thank you again for your time and consideration.
>
> Best regards,
>
> The Authors

---

### Meta-Review · Area_Chair_pXos · 2026-01-06

**Summary:**

This paper received mixed reviews. The main concerns raised in the reviews are:
1. results only demonstrate objects with simple geometry; performance on objects with more complex shapes is questionable (`YmnD`, `Bgc9`).
2. a heavily engineered system relying on existing MLLMs with little conceptual innovation (`YmnD`, `zjTn`, `Bgc9`).
3. limited evaluation for structure alignment through user study metrics (`eGEB`).
4. missing details on correction iterations, failure cases, convergency, and runtime (`eGEB`, `Bgc9`).
5. heavy reliance on a limited training dataset and a primitive library, limiting flexibility (`eGEB`, `Bgc9`).

Overall, some of the concerns have been addressed in the rebuttal. However, I believe major concerns regarding the technical innovation (#2) and evaluation (#3) still remain. Since the overall technical recipe is similar to many existing works, a more thorough and comprehensive evaluation would be necessary to demonstrate superior practicality. However, this is currently lacking. This is a borderline submission and I'm inclined to recommend Reject.

**Reviewer Concerns:**

1. Concern #1 is partially addressed by the 6 new examples with slightly more complex structures in Fig. 18. Nevertheless, a small number of examples is still insufficient to dispel concerns about practical usability.
2. I do not think Concern #2 is sufficiently addressed. I agree with the reviewers that, conceptually, the proposed system follows the same recipe for part-based 3D generation using MLLMs with feedback loops that has been demonstrated in many existing works (including 3DFroMLLM, SINGAPO, and many others).
3. Concern #3 has not been fully addressed. The authors argue that common metrics like LPIPS and SSIM do not accurately reflect structural alignment, which I agree, but this still does not address the fact that the current evaluation is still very limited. Moreover, the paper does not provide details of the human studies, such as the number of participants and the number of sample results, which further undermines the credibility of the human evaluation.
4. Concern #4 is addressed by the additional details in Sec. A.14.
5. Concern #5 is partially addressed by the new examples in Fig. 18, but similarly to Concern #1, it is still not entirely clear how well the system works in practice.

**Reviewer Scores:**

As some of the major concerns remain unresolved, it is unlikely the negative reviewers (`YmnD`, `zjTn`) would increase their ratings based on the current results.

---

### Decision · Program_Chairs · 2026-01-26

Reject